# Rethinking Fair Anomaly Detection From The Group Imbalance Perspective

## Abstract

Anomaly detection (AD) has been widely studied for decades in many real-world applications, including fraud detection in finance, intrusion detection for cyber-security, etc. Existing anomaly detection methods struggle in imbalanced group scenarios, where the unprotected group is significantly larger than the protected group. Specifically, fairness-unaware methods achieve high overall performance by misclassifying more protected group examples as anomalies, while fairness-aware methods overcompensate fairness by labeling excessive unprotected group examples as anomalies, sacrificing overall performance. To address these issues, we propose FADIG, a fairness-aware contrastive learning-based anomaly detection method designed for imbalanced groups. FADIG consists of two key modules: (1) an adaptively re-balanced autoencoder module that dynamically adjusts group contributions to balance fairness with performance and (2) a fairness-aware contrastive learning module that maximizes similarity between protected and unprotected groups to ensure fairness. Moreover, we provide a theoretical analysis showing our proposed contrastive learning regularization guarantees group fairness. Extensive experiments across multiple real-world datasets demonstrate the effectiveness and efficiency of FADIG in achieving both accurate and fair anomaly detection.

## 1 Introduction

Anomaly detection (AD), a.k.a. outlier detection, is referred to as the process of detecting data instances that significantly deviate from the majority of data instances (Chandola et al., 2009). It finds extensive use in a wide variety of applications including financial fraud detection (West & Bhattacharya, 2016; Huang et al., 2018; Rezapour, 2019), pathology analysis in the medical domain (Faust et al., 2018; Shvetsova et al., 2021) and intrusion detection for cybersecurity (Liao et al., 2013; Ahmad et al., 2021; Ahmed et al., 2016). Recent advances in AD primarily focus on learning a scalar anomaly scoring function in an end-to-end fashion (Sohn et al., 2021; Li et al., 2023) or learning patterns of normal examples through a feature extractor (Audibert et al., 2020; Chen et al., 2021; Hou et al., 2021; Yan et al., 2021; Wang et al., 2023). While these approaches are capable of effectively identifying anomalies, they often exhibit significant biases in scenarios of imbalanced groups. In particular, the overrepresentation of an *unprotected* group and underrepresentation of a *protected* group can cause the model to prioritize patterns from the unprotected group, resulting in unfair anomaly predictions.

To mitigate potential bias and ensure fairness for both groups in anomaly detection tasks, several fairness-aware anomaly detection methods have been proposed, including FairOD (Shekhar et al., 2021), TeD-SPAD (Fioresi et al., 2023), DCFOD (Song et al., 2021) and FairSVDD (Zhang & Davidson, 2021). Despite the effectiveness and bias-mitigating capabilities of existing methods, these approaches often overlook the underlying group imbalance that gives rise to such unfairness. As a result, they struggle to maintain fairness and utility simultaneously with imbalanced groups, particularly when the unprotected group is more prevalent than the protected group and the protected group is severely underrepresented.

To illustrate this challenge, we consider the MNIST-USPS dataset (Zhang & Davidson, 2021), where the unprotected group is four times the size of the protected group, and approximately 10% of the total examples are anomalies. Figure 1 and Table 1 present the performance of various anomaly de-

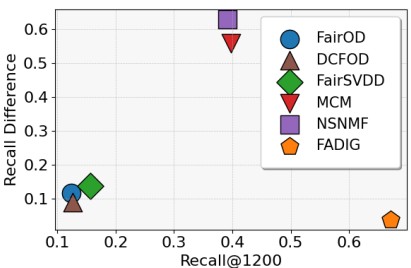

Figure 1: Recall@1200 and the absolute Recall difference of different methods on the MNIST-USPS dataset.

| Method | Unprotected | Protected |
|--------|-------------|-----------|
| FairOD | 117 (984) | 35 (216) |
| DCFOD | 124 (970) | 25 (230) |
| FairSVDD | 141 (741) | 44 (459) |
| MCM | 238 (327) | 234 (873) |
| NSNMF | 196 (294) | 267 (906) |
| FADIG(ours) | 630 (809) | 247 (391) |
| Ground-truth | 882 | 323 |

Table 1: True anomalies out of identified anomalies (number in the parentheses) of different methods in each group.

tection methods on this dataset. In these results, *Recall Diff* refers to the absolute difference in recall between the protected and unprotected groups. Two key observations emerge from the performance of existing anomaly detection methods. First, fairness-unaware methods (e.g., MCM (Yin et al., 2024) and NSNMF (Ahmed et al., 2021)) tend to label more examples from the protected group as anomalies than from the unprotected group (e.g., 873 examples from the protected group labeled as anomalies by MCM versus 327 examples from the unprotected group, as shown in Table 1). While these methods achieve higher overall performance, they exhibit unfair behavior, as evidenced by a high recall difference in Figure 1. This phenomenon occurs because these models prioritize learning frequent patterns from the more abundant unprotected group, while neglecting the under-represented patterns of the protected group. Second, fairness-aware anomaly detection methods (e.g., FairOD, DCFOD, and FairSVDD) often overcompensate by labeling more examples from the unprotected group as anomalies than the actual number of anomalies (e.g., 984 examples labeled as anomalies by FairOD versus 882 actual anomalies in the unprotected group in Table 1). This occurs due to too much attention paid to fairness constraints embedded in their objective functions. Thus, the trade-off between performance and fairness comes at the cost of reduced recall rates, ultimately compromising overall performance for fairness.

These findings highlight a core insight: group imbalance is one root cause of unfairness in anomaly detection. Ignoring this imbalance results in biased representations and unfair predictions; overcorrecting for fairness without modeling such imbalance often results in overall performance loss. A principled approach must therefore reconcile fairness and anomaly detection by explicitly accounting for group imbalance in both representation learning and task objectives.

In this paper, we propose FADIG, a fairness-aware contrastive learning-based anomaly detection method designed from the imbalanced group perspective. FADIG mainly consists of two modules: (1) An **adaptively re-balanced autoencoder module** that introduces learnable weights to automatically balance contributions from protected and unprotected groups, accounting for both normal and anomalous examples, implicitly harmonizing the fairness consideration with the anomaly detection objective; (2) A **fairness-aware contrastive learning module** that aligns representations across groups by maximizing inter-group similarity, reducing group disparity in learned embeddings to ensure fairness. Together, these two components enable FADIG to effectively detect anomalies while promoting group fairness, even under severe imbalance. We also provide a theoretical analysis showing that our proposed method guarantees group fairness. Extensive experiments on multiple real-world datasets demonstrate that FADIG achieves state-of-the-art performance in balancing detection accuracy and fairness. Our contributions are summarized as follows:

- We revisit the problem of fair anomaly detection and identify group imbalance as a key source of unfairness which is largely neglected in existing methods.

- We propose FADIG, a fairness-aware anomaly detection framework that directly addresses group imbalance via fair contrastive representation learning and adaptive objective reweighting.

- Theoretical analysis shows that our proposed method offers group fairness guarantees.

- Extensive empirical studies across multiple real-world datasets validate the effectiveness and efficiency of FADIG.

## 2 PRELIMINARIES

In this paper, we explore the fairness issue in the unsupervised anomaly detection task. Among the various fairness definitions proposed, there is no consensus about the best one to use. In this work, we focus on group fairness notions which usually pursue the equity of certain metrics among the groups. Without loss of generality, we consider the groups here to be the protected group and the unprotected group (e.g., Black and Non-Black in race). We are given a dataset $D = P \cup U$, where $P = \{x_i^P, y_i^P\}_{i=1}^n$ are examples from the protected group, $U = \{x_i^U, y_i^U\}_{i=1}^m$ are examples from the unprotected group, and $x_i^P, x_i^U$ are sampled i.i.d from distributions $\mathcal{P}_P, \mathcal{P}_U$ over the input space $\mathbb{R}^d$ respectively. The ground-truth labels $y_i^P, y_i^U \in \mathcal{Y} = \{0, 1\}$ represent whether the example is an anomaly ($y = 1$) or not, which are given by deterministic labeling functions $a_P, a_U : \mathbb{R}^d \to \mathcal{Y}$, respectively. Note that we do not have access to the labels during training as we focus on the unsupervised anomaly detection setting.

The task of unsupervised anomaly detection is to find a hypothesis $h : \mathbb{R}^d \to \mathcal{Y}$ which identifies a subset $\mathcal{A} \subset D$ whose elements deviate significantly from the normal examples in $D$. This identification is done without the aid of labeled examples, meaning the algorithm must rely on the intrinsic properties of the data, such as distribution, density, or distance metrics, to discern between normal examples and anomalies. The risk of a hypothesis $h$ w.r.t. the true labeling function $a$ under distribution $\mathcal{D}$ using a loss function $\ell : \mathcal{Y} \times \mathcal{Y} \to \mathbb{R}_+$ is defined as: $R_{\mathcal{D}}^\ell(h, a) := \mathbb{E}_{x \sim \mathcal{D}}[\ell(h(x), a(x))]$. We assume that $\ell$ satisfies the triangle inequality. For notation simplicity, we denote $R_P^\ell(h) := R_{\mathcal{P}_P}^\ell(h, a_P)$ and $R_U^\ell(h) := R_{\mathcal{P}_U}^\ell(h, a_U)$. The empirical risks over the protected group $P$ and the unprotected group $U$ are denoted by $\hat{R}_P^\ell$ and $\hat{R}_U^\ell$.

One direction of unsupervised AD is reconstruction-based autoencoder, such as (An & Cho, 2015; Audibert et al., 2020; Hou et al., 2021). Assuming the anomalies possess different features than the normal examples, given an autoencoder over the normal examples, it will be hard to compress and reconstruct the anomalies (Guo et al., 2023; You et al., 2022). The anomaly score can then be defined as the reconstruction loss for each test example. Formally, the autoencoder consists of two main components: an encoder $g_e : \mathbb{R}^d \to \mathbb{R}^r$ and a decoder $g_d : \mathbb{R}^r \to \mathbb{R}^d$, where $r$ is the dimensionality of the hidden representations. $g_e(x)$ encodes the input $x$ to a hidden representation $z$ that preserves the important information of the input. Then, $g_d(z)$ aims to recover $x' \approx x$, a reconstruction of the input from the hidden representation $z$. Overall, the autoencoder can be written as $G = g_d \circ g_e$, i.e. $G(x) = g_d(g_e(x))$. For a given autoencoder-based framework, the anomaly score for $x$ is computed using the reconstruction error as:

$$s(x) = \|x - G(x)\|^2 \tag{1}$$

Throughout the paper, all norms are $\ell_2$ unless otherwise specified. Anomalies tend to exhibit large reconstruction errors because they do not conform to the patterns in the data as coded by the autoencoder. This scoring function is generic in that it applies to a wide range of reconstruction-based AD models, which have different parameterizations of the reconstruction function $G$.

For clarity, we summarize the notations used in the paper in Table 7. In the next section, we introduce our proposed method, which builds upon this autoencoder framework to incorporate fairness.

## 3 PROPOSED METHOD

Our proposed FADIG mainly consists of two modules: an Adaptively Re-balanced Autoencoder Module and a Fairness-aware Contrastive Learning Module.

### 3.1 ADAPTIVELY RE-BALANCED AUTOENCODER

**Re-balanced autoencoder.** As shown in Figure 1 and Table 1 in the Introduction, existing fairness-unaware AD methods overly label more examples from the protected group as anomalies than from the unprotected group, prioritizing learning frequent patterns from the unprotected group while neglecting the under-represented patterns of the protected group. To further investigate the reason, let us first look at the formulation of the reconstruction error for AD methods (Shekhar et al., 2021;

Zhang & Davidson, 2021) with autoencoder-based architecture formulated as follows:

$$\mathcal{L}_{\text{REC}_{\text{unweighted}}} = \sum_{x_i \in P \cup U} \|x_i - G(x_i)\|^2 = \underbrace{\sum_{i=1}^{n} \|x_i^P - G\left(x_i^P\right)\|^2}_{\mathcal{L}_P} + \underbrace{\sum_{i=1}^{m} \|x_i^U - G\left(x_i^U\right)\|^2}_{\mathcal{L}_U}. \tag{2}$$

By decomposing the reconstruction loss into two terms for protected and unprotected groups respectively, we observe that the learning objective in Equation (2) mainly concentrates on learning frequent patterns of the unprotected group (i.e., $\mathcal{L}_U$) due to the more abundant examples in the unprotected group than the protected group in the imbalanced group scenario. As a consequence, it yields higher reconstruction errors for the examples from the protected group and disproportionately labels more examples from the protected group as anomalies than from the unprotected group. Furthermore, when we simply optimize the reconstruction loss in Equation (2) with the fairness loss, the model's capability of learning important frequent patterns is usually diminished by pursuing equal performance for the two groups, as demonstrated in the performance of fair AD methods in Figure 1. This observation inspires us to think about whether we can ensure fairness while maintaining good performance under group imbalance.

To answer this question, we propose to reweigh the reconstruction losses during training to assign different emphasizes to each group. Specifically, we design a re-balanced autoencoder by minimizing the reweighted reconstruction loss as follows:

$$\mathcal{L}_{\text{REC}} = (1 - \epsilon)\mathcal{L}_U + \epsilon\mathcal{L}_P, \tag{3}$$

where $\epsilon$ is a weight to adjust trade-off between learning frequent and under-represented patterns. Notice that if we set $\epsilon$ as a hyperparameter, it would be non-trivial to determine the appropriate range of $\epsilon$. Next, we introduce our hyperparameter-free and parameter-free adaptive design of $\epsilon$ which dynamically adjusts the trade-off by fitting normal examples from both protected and unprotected groups at each training iteration.

**Adaptive weight $\epsilon$.** We theoretically characterize a range of $\epsilon$ that promotes fairness and accordingly propose an adaptive design of $\epsilon$ that is fully computed from data without any hyperparameters. In anomaly detection research based on autoencoders, a common assumption is that the model can effectively reconstruct normal samples well but struggles to reconstruct anomalies (Guo et al., 2023; You et al., 2022; Deng & Li, 2022). As a result, inputs with high reconstruction errors are classified as anomalies. Consider the four subgroups of data samples in the fair anomaly detection task: unprotected/protected normal examples (UN/PN) and unprotected/protected anomalies (UA/PA). Ideally, the model should only fit normal samples, *i.e.*, the two subgroups, UN and PN. We assume that the model is capable of fitting two out of the four subgroups. Let $\mathcal{L}_0^t$ denote the loss of the unfitted model on the subgroup $t \in \{\text{UN, PN, UA, PA}\}$, and let $\mathcal{L}_1^t$ denote the loss of the fitted model on the subgroup $t$. $\Delta^t = \mathcal{L}_0^t - \mathcal{L}_1^t > 0$ means the difference of loss between the fitted model and the unfitted one on the subgroup $t$. For the design of $\epsilon$, we have the following lemma:

**Lemma 3.1.** *A proper weight $\epsilon$ for model fitting on normal examples in both protected and unprotected groups should be within the range $\frac{\Delta^{UA}}{\Delta^{UA} + \Delta^{PN}} < \epsilon < \frac{\Delta^{UN}}{\Delta^{UN} + \Delta^{PA}}$ such that fitting normal samples of both groups leads to a lower loss compared to fitting abnormal samples from either group. Although $\frac{\Delta^{UA}}{\Delta^{UA} + \Delta^{PN}}$ and $\frac{\Delta^{UN}}{\Delta^{UN} + \Delta^{PA}}$ are unknown, the following $\epsilon$ always lies in the desired range:*

$$\epsilon = \frac{\mathcal{L}_0^U - \mathcal{L}_U}{\mathcal{L}_0^U - \mathcal{L}_U + \mathcal{L}_0^P - \mathcal{L}_P} \tag{4}$$

*where $\mathcal{L}_0^U = \mathcal{L}_0^{UN} + \mathcal{L}_0^{UA}$ and $\mathcal{L}_0^P = \mathcal{L}_0^{PN} + \mathcal{L}_0^{PA}$.*

We provide the proof of Lemma 3.1 in Appendix E.1. In our implementation, we estimate $\mathcal{L}_0^U = \sum_{i \in U} \|x_i - \overline{G_U(x)}\|^2$, where $\overline{G_U(x)} = \frac{1}{|U|} \sum_{i \in U} G(x_i)$, and $\mathcal{L}_0^P = \sum_{i \in P} \|x_i - \overline{G_P(x)}\|^2$, where $\overline{G_P(x)} = \frac{1}{|P|} \sum_{i \in P} G(x_i)$. Other estimators of $\mathcal{L}_0^U$ and $\mathcal{L}_0^P$ are discussed in Appendix E.2.

**Remark:** According to the design of $\epsilon$ in Lemma 3.1, when FADIG fails to effectively learn frequent patterns of the unprotected group, the loss $\mathcal{L}_U$ becomes large and $\epsilon$ decreases accordingly, thereby assigning a larger weight to $\mathcal{L}_U$. This adjustment encourages the model to focus more on learning frequent patterns of unprotected group in subsequent iterations. Conversely, if the model

overlooks under-represented patterns (*i.e.*, protected group), their corresponding weight $\epsilon$ increases, compelling the model to pay more attention to learn under-represented patterns and thus ensuring fairness. Overall, the adaptive nature of $\epsilon$ dynamically prioritizes the optimization of the group that is less well-fitted during training, thus harmonizing anomaly detection with fairness.

### 3.2 FAIRNESS-AWARE CONTRASTIVE LEARNING

Aside from adaptively rebalancing group contributions in the training loss, we further explore how to improve representation learning under group imbalance. Existing anomaly detection models (Song et al., 2021; Zhang & Davidson, 2021; Fioresi et al., 2023) typically focus on modeling frequent patterns dominated by the unprotected group while under-represented patterns associated with the protected group are inadequately captured, resulting in misclassification and unfair outcomes.

Contrastive loss has demonstrated strong benefits in learning robust and transferable representations across diverse groups, domains, and classes (Sohn et al., 2021; Chen et al., 2020). It operates by pulling similar samples closer in the embedding space while pushing dissimilar ones apart with data augmentation techniques (Khosla et al., 2020; Zhang et al., 2023). Inspired by this, we propose a fairness-aware contrastive loss tailored for anomaly detection. Specifically, rather than relying on data augmentations, we maximize the cosine similarity between representations of samples from the protected and unprotected groups to directly encourage fair and shared pattern learning. At the same time, to prevent representation collapse (Wang & Isola, 2020) and preserve anomaly discrimination, we promote intra-group uniformity by pushing apart representations within each group. Formally, our fairness-aware contrastive loss is defined as:

$$\mathcal{L}_{\text{FAC}} = \underbrace{-\log\left(\frac{\sum_j \sum_k \text{sim}\left(z_j^P, z_k^U\right)}{mn}\right)}_{\mathcal{L}_{\text{fair}}} + \underbrace{\log\left(\frac{\sum_{j\neq k} \text{sim}\left(z_j^U, z_k^U\right)}{m(m-1)} + \frac{\sum_{j\neq k} \text{sim}\left(z_j^P, z_k^P\right)}{n(n-1)}\right)}_{\mathcal{L}_{\text{unif}}} \tag{5}$$

Following the interpretation of contrastive loss in (Wang & Isola, 2020), the first term $\mathcal{L}_{\text{fair}}$ promotes alignment across groups, encouraging the model to learn shared representations for both protected and unprotected groups to ensure fairness. The second term $\mathcal{L}_{\text{unif}}$ encourages the uniformity of the representations within each group, improving the model's ability to separate normal and anomalies and ensuring anomaly detection performance.

Next, we theoretically show that the proposed fairness-aware contrastive loss $\mathcal{L}_{\text{FAC}}$ offers group fairness guarantees by bounding the risk difference.

### 3.3 FAIRNESS ANALYSIS

In this subsection, we analyze how $\mathcal{L}_{\text{FAC}}$ promotes group fairness. We begin by introducing the definition of $f$-divergence, which enables us to derive an upper bound on the group risk difference.

**Definition 3.2.** ($f$-divergence (Ali & Silvey, 1966) ) Let $P$ and $Q$ be two distribution functions with densities $p$ and $q$, respectively. Let $p$ be absolutely continuous w.r.t $q$ and both be absolutely continuous with respect to a base measure $dx$. Let $f : \mathbb{R}_+ \to \mathbb{R}$ be a convex, lower semi-continuous function that satisfies $f(1) = 0$. The $f$-divergence $D_f$ is defined as:

$$D_f(P \parallel Q) = \int q(x) f\left(\frac{p(x)}{q(x)}\right) dx. \tag{6}$$

Many commonly used divergences in machine learning, such as KL divergence and total variation distance, are special cases of $f$-divergence. Table 8 in Appendix D provides examples. Following (Nguyen et al., 2010), $f$-divergence can be estimated via a variational formulation:

$$D_f(P \parallel Q) \geq \sup_{T \in \mathcal{T}} \mathbb{E}_{x \sim P}[T(x)] - \mathbb{E}_{x \sim Q}[f^*(T(x))] \tag{7}$$

where $f^*$ is the (Fenchel) conjugate function of $f$ defined as $f^*(y) := \sup_{x \in \mathbb{R}_+}\{xy - f(x)\}$, $T : \mathcal{X} \to \text{dom } f^*$, and $\mathcal{T}$ is the set of all measurable functions.

Next, with the help of Rademacher complexity (Shalev-Shwartz & Ben-David, 2014) (detailed definition provided in Appendix C), we now provide a fairness bound for the performance difference between the protected and unprotected groups $R_P^\ell(h) - R_U^\ell(h)$:

Table 2: Characteristics of datasets.

| Datasets | Unprotected Group | | Protected Group | | #Features | Sensitive Attribute | Anomaly Definition |
|---|---|---|---|---|---|---|---|
| | #Instances | #Anomaly | #Instances | #Anomaly | | | |
| MNIST-USPS | 7,785 | 882 | 1,876 | 323 | 1,024 | Source of the digits | Digit 0 or not |
| MNIST-Invert | 7,344 | 441 | 408 | 38 | 1,024 | Color of the digits | Digit 0 or not |
| COMPAS | 1,839 | 325 | 299 | 39 | 8 | Race | Reoffending or not |
| CelebA | 41,919 | 4,008 | 7,300 | 1,142 | 39 | Gender | Attractive or not |

**Theorem 3.3.** *(Fairness with Rademacher Complexity) Suppose $\ell : \mathcal{Y} \times \mathcal{Y} \to [0,1]$, $f^*$ is L-Lipschitz continuous, and $[0,1] \subseteq \operatorname{dom} f^*$. Let $U$ and $P$ be two empirical distributions corresponding to datasets containing $m$ and $n$ data points sampled i.i.d. from $P_U$ and $P_P$, respectively. Let us denote $\mathfrak{R}$ as the Rademacher complexity of a given hypothesis class, and define $\ell \circ \mathcal{H} := \{x \mapsto \ell(h(x), h'(x)) : h, h' \in \mathcal{H}\}$. Let $h^*$ be the ideal joint hypothesis, i.e., $h^* = \arg\min_{h \in \mathcal{H}} R_U^\ell(h) + R_P^\ell(h)$. For any $\delta \in (0,1)$, with probability at least $1 - \delta$, we have:*

$$R_P^\ell(h) - R_U^\ell(h) \le D_f(U \| P) + \hat{R}_U^\ell(h^*) + \hat{R}_P^\ell(h^*) + 4\mathfrak{R}_U(\ell \circ \mathcal{H}) + 2(L+1)\mathfrak{R}_P(\ell \circ \mathcal{H}) + 2\sqrt{\frac{\log \frac{1}{\delta}}{2m}} + 2\sqrt{\frac{\log \frac{1}{\delta}}{2n}}$$

Under the assumption of an ideal joint hypothesis $h^*$, fairness (*i.e.*, the risk difference between the protected and unprotected groups) can be improved by minimizing the discrepancy between the hidden representation of the examples from two groups and regularizing the model to limit the complexity of the hypothesis class. In Appendix E.3, we provide a full proof of Theorem 3.3 and analyze how minimizing the objective $\mathcal{L}_{\text{FAC}}$ leads to small empirical $f$-divergence $D_f(U \| P)$ for total variation, and thereby promotes fairness.

**Summary of Our Method.** To summarize, the overall training scheme of FADIG is to minimize:

$$\mathcal{L}_{\text{overall}} = \mathcal{L}_{\text{REC}} + \alpha \mathcal{L}_{\text{FAC}},$$

where $\alpha$ is a hyperparameter to balance the reconstruction loss and the contrastive loss. During the inference stage, we compute the reconstruction error for each instance and rank them. The top $k$ examples with the highest reconstruction errors are flagged as anomalies. While our method focuses on the binary group case, we can naturally extend it to the multi-value multi-group case as discussed in Appendix E.4.

## 4 EXPERIMENTS

In this section, we experimentally analyze and compare our proposed FADIG with other anomaly detection methods. We try to answer the following research questions:

- Q1: How does FADIG compare with other baselines on imbalanced datasets?
- Q2: How does FADIG perform with different imbalance ratios of the two groups?
- Q3: How does each module contribute to FADIG?

### 4.1 EXPERIMENTAL SETUP

**Datasets:** We conduct experiments on two image datasets, MNIST-USPS and MNIST-Invert (Zhang & Davidson, 2021), and two tabular datasets, COMPAS (Angwin et al., 2022) and CelebA (Liu et al., 2015). The detailed characteristics of the datasets are provided in Table 2.

**Baseline Methods:** In our experiments, we compare our proposed framework FADIG with the following fairness-aware anomaly detection baselines: (1) **FairOD** (Shekhar et al., 2021), a fair AD method which incorporates various group fairness criteria including flag rate parity, statistical parity and group fidelity into its training; (2) **DCFOD** (Song et al., 2021), a fair deep clustering-based method, which leverages deep clustering to discover the intrinsic cluster structure and out-of-structure instances; (3) **FairSVDD** (Zhang & Davidson, 2021), an adversarial network to decorrelate the relationships between sensitive attributes and the learned representations. We also compare with the following fairness-agnostic AD baselines: (4) **MCM** (Yin et al., 2024), a masked modeling method to address AD by capturing intrinsic correlations between features in the training set; (5) **NSNMF** (Ahmed et al., 2021), a non-negative matrix factorization method, which incorporates the neighborhood structural similarity information to improve the anomaly detection performance; (6) **ReContrast** (Guo et al., 2023), a reconstructive contrastive learning-based method for

Table 3: Performance on image datasets. The best score is marked in bold.

| Methods | MNIST-USPS (K=1200) | | | | MNIST-Invert (K=500) | | | |
|---|---|---|---|---|---|---|---|---|
| | Recall@K | ROCAUC | Rec Diff | Time(s) | Recall@K | ROCAUC | Rec Diff | Time(s) |
| FairOD | 12.35±1.13 | 50.00±0.28 | 11.56±0.64 | 29.57 | 7.52±0.74 | 50.40±0.20 | 8.26±1.27 | 20.25 |
| DCFOD | 12.63±0.33 | 50.09±0.27 | 8.99±0.83 | 710.33 | 6.95±0.91 | 50.54±0.54 | **7.23±2.02** | 1277.31 |
| FairSVDD | 15.62±1.52 | 58.33±1.18 | 13.75±2.56 | 768.79 | 12.41±0.76 | 49.67±3.98 | 12.46±2.12 | 843.12 |
| MCM | 39.75±0.23 | 78.80±1.02 | 55.81±0.80 | 417.09 | 25.35±0.56 | 80.96±0.49 | 80.13±1.41 | 752.36 |
| NSNMF | 39.16±0.84 | 65.38±0.58 | 62.90±3.84 | 28.53 | 51.79±0.61 | 74.21±0.34 | 51.07±1.79 | 18.97 |
| ReContrast | 64.29±3.18 | 83.46±3.77 | 41.16±5.63 | 116.75 | 64.22±1.60 | 85.13±5.19 | 56.50±11.23 | 117.15 |
| FADIG | **67.19±0.33** | **91.28±0.46** | **3.77±2.18** | 121.97 | **71.82±0.63** | **97.99±0.07** | 9.78±3.10 | 60.42 |

Table 4: Performance on tabular datasets. The best score is marked in bold.

| Methods | COMPAS (K=350) | | | | CelebA (K=5000) | | | |
|---|---|---|---|---|---|---|---|---|
| | Recall@K | ROCAUC | Rec Diff | Time(s) | Recall@K | ROCAUC | Rec Diff | Time(s) |
| FairOD | 16.56±2.12 | 50.09±1.28 | 7.97±1.23 | 4.18 | 8.93±0.14 | 49.94±0.12 | **0.68±0.56** | 78.92 |
| DCFOD | 16.08±1.94 | 49.55±1.21 | 9.81±1.76 | 115.86 | 9.66±0.69 | 49.92±0.14 | 7.83±1.26 | 2517.68 |
| FairSVDD | 15.33±2.10 | 52.68±5.29 | 11.57±4.06 | 6.81 | 10.19±0.50 | 58.40±1.02 | 10.95±1.93 | 243.17 |
| MCM | 21.10±0.54 | 50.97±0.43 | 6.29±2.66 | 38.12 | 11.03±0.38 | 46.23±3.46 | 26.15±9.31 | 640.12 |
| NSNMF | 22.92±0.32 | 57.97±0.66 | 36.78±1.71 | 7.69 | 10.91±0.54 | 50.45±0.30 | 8.04±1.33 | 1927.55 |
| FADIG | **34.38±0.36** | **61.45±0.47** | **5.97±4.34** | 19.88 | **11.96±0.49** | **59.43±0.42** | 4.72±1.26 | 48.93 |

domain-specific anomaly detection. Notice that as ReContrast is designed for image data, we only evaluate it on MNIST-USPS and MNIST-Invert datasets.

**Metrics:** To measure the model performance and group fairness, we choose three widely-used metrics (Shekhar et al., 2021; Zhang & Davidson, 2021; Ahmed et al., 2021): (1) **Recall@K**, which measures the proportion of anomalies found in the top-k recommendations; (2) **ROCAUC**, which computes the area under the receiver operating characteristic curve; (3) **Rec Diff**, which measures the absolute value of the recall difference between two groups. To provide a more comprehensive assessment, we include results over additional metrics in Appendix F.3.

**Training details:** For the COMPAS dataset, we use a two-layer MLP with hidden units of [32, 32]. For all the other datasets, we use MLP with one hidden layer of dimension 128. We set the hyperparameter $\alpha = 4$ across all the data sets. We include the results with different choices of $\alpha$ in Appendix F.8 and find that FADIG is robust to the choice of $\alpha$. All our experiments were executed using one Tesla V100 SXM2 GPUs, supported by a 12-core CPU operating at 2.2GHz. We provide more implementation details in Appendix F.1.

**Additional results**, such as comparisons with variants of alternative reweighting heuristics, are provided in Appendix F.4, and results across different anomaly types are discussed in Appendix F.6. We also discuss the limitations and broader impact of our work in Appendix G.

### 4.2 EFFECTIVENESS AND EFFICIENCY OF FADIG (Q1)

We first evaluate the effectiveness and efficiency of FADIG through comparison with baselines across four datasets by four independent runs. The task performance (*i.e.*, Recall@$K$ and RO-CAUC), group fairness measure (*i.e.*, Rec Diff), and their average training time are presented in Tables 3 and 4 (see Appendix F.2 for results with different $K$). We can observe that the fair AD baselines (FairOD, DCFOD, and FairSVDD) typically exhibit low discrepancies in recall. However, they also tend to suffer from reduced recall rates and ROCAUC scores, suggesting a compromise in overall task performance to enhance fairness. On the other hand, the baselines that do not account for fairness, including MCM, NSNMF, and ReContrast, demonstrate high recall rates and ROCAUC scores but often at the expense of fairness, as evidenced by significant disparities across groups (*i.e.*, a higher Rec Diff). In contrast, FADIG not only excels in task performance but also promotes fairness, underscoring the effectiveness of our design in harmonizing fairness with anomaly detection in group imbalance. Additionally, the training time of FADIG is always among the top 4 fastest methods across different datasets, showing the efficiency of our method. Due to limited space, we present comparisons with more baselines on additional large-scale datasets in Appendix F.5 and F.7.

Table 5: Performance on MNIST-USPS with different ratios. The best score is marked in bold.

| Methods | $\lvert U \rvert : \lvert P \rvert = 1 : 1$ (K=650) | | | $\lvert U \rvert : \lvert P \rvert = 2 : 1$ (K=1000) | | | $\lvert U \rvert : \lvert P \rvert = 4 : 1$ (K=1200) | | |
|---|---|---|---|---|---|---|---|---|---|
| | Recall@K | ROCAUC | Rec Diff | Recall@K | ROCAUC | Rec Diff | Recall@K | ROCAUC | Rec Diff |
| FairOD | 17.52±1.17 | 50.13±0.64 | 2.14±0.62 | 17.30±1.24 | 49.73±0.74 | 5.11±0.55 | 13.61±0.22 | 50.22±0.13 | 10.58±1.01 |
| DCFOD | 17.08±0.50 | 50.09±0.30 | 3.25±0.94 | 16.92±0.81 | 49.54±0.42 | 2.76±0.51 | 14.14±1.03 | 50.44±0.60 | 7.11±0.83 |
| FairSVDD | 24.56±2.95 | 54.87±3.36 | 14.24±7.90 | 18.09±3.46 | 52.77±1.72 | 4.85±3.75 | 21.10±2.79 | 63.46±9.56 | 18.38±4.91 |
| MCM | 52.22±1.35 | 74.62±1.24 | 17.13±2.73 | 53.63±1.76 | 76.80±1.04 | 8.17±6.36 | 41.99±4.06 | 74.09±0.45 | 22.85±4.60 |
| NSNMF | 48.71±0.39 | 68.96±0.24 | 40.25±2.17 | 41.07±2.77 | 64.08±1.67 | 54.18±3.11 | 38.87±1.09 | 64.71±0.63 | 62.98±1.47 |
| ReContrast | 45.92±1.85 | 80.17±3.08 | 42.52±3.31 | 51.39±1.75 | 83.13±2.94 | 26.16±1.79 | 57.69±2.36 | 79.17±4.09 | 20.69±3.57 |
| FADIG | **65.58±0.47** | **85.38±0.37** | **0.93±0.87** | **66.84±0.83** | **89.17±0.09** | **2.32±1.08** | **66.63±0.72** | **90.15±0.22** | **1.84±0.68** |

Table 6: Performance on COMPAS dataset with different ratios. The best score is marked in bold.

| Methods | $\lvert U \rvert : \lvert P \rvert = 1 : 1$ (K=80) | | | $\lvert U \rvert : \lvert P \rvert = 2 : 1$ (K=120) | | | $\lvert U \rvert : \lvert P \rvert = 5 : 1$ (K=240) | | |
|---|---|---|---|---|---|---|---|---|---|
| | Recall@K | ROCAUC | Rec Diff | Recall@K | ROCAUC | Rec Diff | Recall@K | ROCAUC | Rec Diff |
| FairOD | 13.68±2.67 | 50.10±0.85 | 11.97±1.48 | 13.11±0.50 | 50.11±0.74 | 6.60±0.97 | 12.54±1.37 | 49.58±0.87 | 7.68±0.72 |
| DCFOD | 11.54±4.62 | 48.50±2.69 | 7.69±4.445 | 15.95±3.00 | 53.28±0.75 | 10.68±2.67 | 12.96±2.02 | 49.76±1.16 | 6.36±0.70 |
| FairSVDD | 16.24±2.18 | 52.34±1.38 | 6.84±3.20 | 14.53±1.84 | 51.69±2.15 | 7.69±3.77 | 14.10±4.53 | 50.04±4.98 | 14.87±7.54 |
| MCM | 18.38±0.60 | 40.77±0.25 | 7.69±3.63 | 16.24±0.01 | 40.42±0.12 | 10.26±4.80 | 18.81±0.60 | 44.04±0.15 | 5.76±2.31 |
| NSNMF | 20.08±0.74 | 53.86±0.42 | 14.53±10.36 | 19.09±1.31 | 53.28±0.75 | 10.68±2.67 | 20.09±2.22 | 53.86±1.28 | 10.77±5.40 |
| FADIG | **29.91±0.74** | **61.87±1.89** | **3.42±1.48** | **28.42±0.43** | **57.39±2.84** | **1.92±1.72** | **29.77±1.31** | **58.05±1.34** | **4.83±0.78** |

## 4.3 DATA IMBALANCE STUDY (Q2)

To further study the performance of FADIG in handling imbalanced group, we vary the levels of group imbalance within the image dataset MNIST-USPS and the tabular dataset COMPAS. We report the average results of four independent runs in Tables 5 and 6. By observation, we find that most fair AD methods (e.g., FairOD, DCFOD and FairSVDD) tend to suffer from the imbalanced groups scenario when the unprotected group gradually becomes dominant (e.g., $\lvert U \rvert : \lvert P \rvert = 4 : 1$ for MNIST-USPS dataset and $\lvert U \rvert : \lvert P \rvert = 5 : 1$ for COMPAS dataset). Specifically, FairOD achieves 17.52% recall rate when $\lvert U \rvert : \lvert P \rvert = 1 : 1$ and its performance decreases to 13.61% when $\lvert U \rvert : \lvert P \rvert = 4 : 1$. We attribute this phenomenon to too much attention paid to fairness constraints embedded in their objective functions and these methods ultimately compromise overall performance for fairness. Different from these fair AD methods, FADIG consistently outperforms the baselines in terms of both task efficacy and fairness across different group ratios. The advantages of using FADIG become more pronounced with increasing level of group imbalance. For instance, while the performance of fair AD baselines drops with higher imbalance ratios on MNIST-USPS, FADIG adeptly sustains superior task performance alongside enhanced fairness, showcasing its robustness against data imbalance. This verifies the effectiveness of adaptively re-balanced autoencoder by learning both frequent and under-represented patterns from two groups.

## 4.4 ABLATION STUDY (Q3)

To assess the contribution of each component in FADIG, we conduct an ablation study to demonstrate the necessity of each component of FADIG on the MNIST-USPS and COMPAS datasets. The experimental results are presented in Figure 2. Specifically, FADIG-W refers to a variant of our method replacing the re-balancing autoencoder with the unweighted reconstruction loss in Equation (2); FADIG-F and FADIG-U remove $\mathcal{L}_{\text{fair}}$ and $\mathcal{L}_{\text{unif}}$ in Equation (5), respectively; FADIG-C substitutes the proposed fair contrastive loss $\mathcal{L}_{\text{FAC}}$ with the traditional contrastive loss (Chen et al., 2020); FADIG-R removes the entire $\mathcal{L}_{\text{FAC}}$. We have the following observations: (1) FADIG greatly outperforms FADIG-F and FADIG-U, highlighting the importance of both $\mathcal{L}_{\text{fair}}$ and $\mathcal{L}_{\text{unif}}$ in achieving accurate and fair anomaly detection. (2) While FADIG-C can sometimes achieve comparable recall to FADIG, it consistently yields a higher recall difference between groups. This suggests that simply using a standard contrastive loss is insufficient for promoting group fairness, validating the need for our fairness-aware regularization. (3) Compared with FADIG, FADIG-W has a lower recall rate and a higher recall difference. It suggests that classical unweighted reconstruction loss tends to mainly focus on learning the frequent patterns of the unprotected group while ignoring the protected group, and validates the effectiveness of our designed re-balanced autoencoder. (4) While FADIG-R has relatively high recall rates and low recall differences on the two datasets, FADIG still surpasses it in all metrics, demonstrating the necessity of $\mathcal{L}_{\text{FAC}}$.

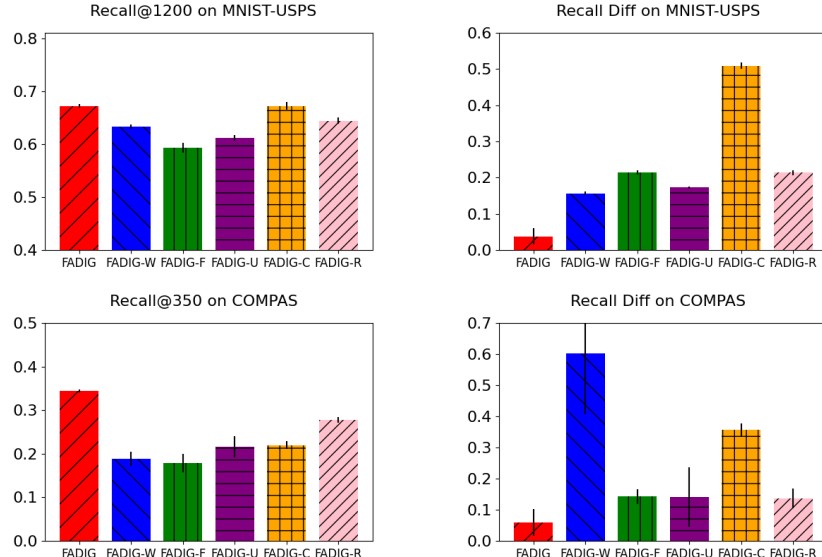

Figure 2: Ablation Study of variants of FADIG on MNIST-USPS and COMPAS datasets.

## 5 RELATED WORK

In this section, we briefly review the related work on unsupervised anomaly detection and fair machine learning. A more extensive discussion is provided in Appendix B.

**Unsupervised Anomaly Detection** does not require labeled data and has been widely applied in domains such as cybersecurity, healthcare, and finance (West & Bhattacharya, 2016; Faust et al., 2018; Shvetsova et al., 2021; Ahmad et al., 2021). As surveyed in Pang et al. (2022), most approaches either learn representations of normal patterns using feature extractors (Audibert et al., 2020; Chen et al., 2021; Hou et al., 2021; Yan et al., 2021; Wang et al., 2023), or directly learn scalar anomaly scores in an end-to-end fashion (Sohn et al., 2021; Li et al., 2023; Jiang et al., 2022). Our work extends autoencoder-based reconstruction models, but crucially introduces two fairness-driven innovations: an adaptive reweighting mechanism that balances groups without additional parameters or hyperparameters, and a contrastive objective that directly promotes shared representations across groups without relying on conventional data augmentations.

**Fair Machine Learning** aims to mitigate bias and ensure equitable outcomes with respect to protected attributes (Zemel et al., 2013; Bolukbasi et al., 2016; Hashimoto et al., 2018; Zhang et al., 2018; Kobren et al., 2019). Within anomaly detection, recent works have begun to explore fairness (Shekhar et al., 2021; Zhang et al., 2023; Yin et al., 2024). However, these methods often overlook the central role of group imbalance, leading to persistent representation disparity and unfair detection performance. We identify group imbalance as a critical yet largely ignored source of unfairness in AD, and explicitly address it through two novel methodological components: a provably fair reweighted autoencoder and a fairness-driven contrastive loss that jointly enforce group-balanced training and fair representation learning. We also provide theoretical guarantees for fairness in both components, closing a gap in the existing literature.

## 6 CONCLUSION

We propose FADIG, a fairness-aware anomaly detection framework tailored for imbalanced group scenarios. FADIG incorporates a fairness-aware contrastive learning module that promotes representation similarity between protected and unprotected groups to enhance group fairness. To balance detection performance, we introduce a hyperparameter-free and parameter-free adaptively reweighted autoencoder that dynamically adjusts the contribution of each group during training with provable guarantees. We further provide a theoretical guarantee by deriving an upper bound on group risk disparity, showing that our contrastive regularization promotes group fairness. Extensive experiments across multiple real-world datasets demonstrate the effectiveness and efficiency of FADIG in achieving both fair and accurate anomaly detection.

## 7 STATEMENTS

For Reproducibility, in Appendix F.1 we present the experimental setup, training details, and implementation code. In Appendix E, we include the proofs for all the theoretical results. Regarding LLM usage, we use LLMs lightly only to polish the writing.

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

## A RELATED WORK

### A.1 UNSUPERVISED ANOMALY DETECTION

Anomaly detection (AD) has been studied for decades across applications such as fraud detection (West & Bhattacharya, 2016; Huang et al., 2018), medical diagnosis (Faust et al., 2018; Shvetsova et al., 2021), intrusion detection (Liao et al., 2013; Ahmad et al., 2021), and fault detection in safety-critical systems (Ju et al., 2021). Given various types of anomalies (Breunig et al., 2000; Kriegel et al., 2009; Bouman et al., 2024), the authors of Pang et al. (2022) divide the existing anomaly detection methods into two major branches. The first branch focuses on learning representations of normal samples via feature extractors (Audibert et al., 2020; Chen et al., 2021; Hou et al., 2021; Yan et al., 2021; Wang et al., 2023). For example, Audibert et al. (2020) train an encoder–decoder to amplify reconstruction error through adversarial training, while Chen et al. (2021) propose a GAN-based autoencoder to model multivariate time series. Our method also builds on autoencoder-based reconstruction, but introduces an adaptive reweighting scheme to balance groups and a fairness-aware contrastive regularizer to reduce representation disparity between groups. The second branch aims at learning scalar anomaly scores in an end-to-end fashion (Sohn et al., 2021; Li et al., 2023; Jiang et al., 2022). Notably, the authors of Sohn et al. (2021) combine distribution-augmented contrastive regularization with a one-class classifier to detect anomalies. While Sohn et al. (2021) uses image augmentations, such as rotations, to form positive pairs and negative pairs for contrastive learning, we instead use existing examples for contrastive learning and thus our method is applicable to various types of data, not limited to image data. To be more specific, we design a fairness-aware contrastive learning loss which minimizes the representation disparity of the groups for fairness, and encourages the uniformity within each group for better anomaly detection.

## B RELATED WORK

### B.1 FAIR MACHINE LEARNING

Fair Machine Learning aims to amend the biased machine learning models to be fair or invariant regarding specific variables. A surge of research in fair machine learning has been done in the machine learning community (Kobren et al., 2019; Zemel et al., 2013; Bolukbasi et al., 2016; Hashimoto et al., 2018; Zhang et al., 2018; Park et al., 2022). For example, Zemel et al. (2013) presents a learning algorithm for fair classification by enforcing group fairness and individual fairness in the obtained data representation; Hashimoto et al. (2018) develop a robust optimization framework that minimizes the worst case risk over all distributions and preserves the minority group in an imbalanced data set; Zhang et al. (2018) present an adversarial-learning based framework for mitigating the undesired bias in modern machine learning models. Park et al. (2022) propose a fair supervised contrastive loss to train a fair representation model. However, they rely on target labels and need negative samples since their method is based on supervised contrastive learning. Instead, our designed fair contrastive learning loss uses examples from different groups in an unsupervised way to minimize the representation disparity of the groups and encourage the uniformity within each group.

Several studies investigate fairness–utility trade-offs using multi-objective optimization (Badar et al., 2024; Liu & Vicente, 2022; Martinez et al., 2020; Wang et al., 2021). These works characterize Pareto-efficient trade-offs between accuracy and fairness across various tasks. While we do not explicitly optimize Pareto fronts, our method harmonizes fairness and utility in the unsupervised AD setting by automatically rebalancing groups and aligning their latent representations.

In the specific field of fair anomaly detection, Zhang & Davidson (2021) utilizes the adversarial generative nets to ensure group fairness and use one-class classification to detect the anomalies; Song et al. (2021) introduces fairness adversarial training and proposes a dynamic weight to reduce the negative impacts from outlier points. However, these existing fair anomaly detection methods (Song et al., 2021; Zhang & Davidson, 2021; Fioresi et al., 2023) tend to suffer from the representation disparity issue when groups are imbalanced. Our approach directly addresses this by combining (i) a hyperparameter-free, parameter-free reweighted autoencoder that adaptively balances groups with theoretical guarantees, and (ii) a fairness-driven contrastive objective that maximizes similarity between protected and unprotected groups while maintaining within-group uniformity.

Table 7: Notation Table.

| Symbol | Description |
|---|---|
| $x$ | input feature |
| $\mathcal{P}_P$ | Protected group's distribution |
| $\mathcal{P}_U$ | Unprotected group's distribution |
| $P$ | Protected group's empirical distribution |
| $U$ | Unprotected group's empirical distribution |
| $n/m$ | Size of protected/unprotected group |
| $a_P/a_U$ | labeling functions on protected/unprotected group |
| $\ell$ | Loss function |
| $R_D^\ell(h)$ | Risk of hypothesis $h$ over data $D$ |
| $\hat{R}_D^\ell(h)$ | Empirical risk of hypothesis $h$ over data $D$ |
| $s(x)$ | Anomaly score of example $x$ |
| $\mathfrak{R}_D(\mathcal{F})$ | Rademacher complexity of $\mathcal{F}$ given data $D$ |
| $D_f(P \parallel Q)$ | f-divergence between distributions $P$ and $Q$ |

While the techniques of loss reweighting and contrastive learning have been extensively studied, we contribute to the novel design of both modules through the lens of fairness under imbalance. For loss reweighting, most existing loss reweighting methods (Song et al., 2021; Zhang & Davidson, 2021) introduce a learnable vector to learn the sample-wise weight or introduce hyperparameters for samples from different unprotected and protected groups. Our proposed method presents a hyperparameter-free and parameter-free reweighting scheme by automatically and adaptively weighing the importance of unprotected and protected groups with a theoretical guarantee. Compared with the existing reweighting methods, our method either reduces the number of parameters in the model or gets rid of the hyperparameter for group weight. For contrastive learning, we design the contrastive learning loss from the fairness perspective. The traditional contrastive learning loss uses two augmentation methods to generate two views to form the positive pairs, maximizes the similarity of positive pairs, and minimizes the similarity of negative pairs. Most of the existing fairness methods (Wang et al., 2022; Zhang et al., 2022; Chen et al., 2025) adopt this procedure, which largely rely on augmentations and do not directly encourage fairness. In our method, we maximize the similarity between representations of samples from the protected and unprotected groups (instead of two augmented views) to directly encourage fair and shared pattern learning. Furthermore, our design alleviates the impact of the imbalanced groups.

## C  RADEMACHER COMPLEXITY

The Rademacher complexity for a function class is:

**Definition C.1.** (Rademacher Complexity (Shalev-Shwartz & Ben-David, 2014)) Given a space $\mathcal{X}$, and a set of i.i.d. examples D $= \{x_1, x_2, ..., x_{|D|}\} \subseteq \mathcal{X}$, for a function class $\mathcal{F}$ where each function $r : \mathcal{X} \to \mathbb{R}$, the empirical Rademacher complexity of $\mathcal{F}$ is given by:

$$\mathfrak{R}_D(\mathcal{F}) = \mathbb{E}_\sigma \left[ \sup_{r \in \mathcal{F}} \left( \frac{1}{|D|} \sum_{i=1}^{|D|} \sigma_i r(z_i) \right) \right] \tag{8}$$

Here, $\sigma_1, ..., \sigma_m$ are independent random variables uniformly drawn from $\{-1, 1\}$.

Recall that we previously defined $R_{\mathcal{D}}^\ell(h) \coloneqq R_{\mathcal{D}}^\ell(h, a) = \mathbb{E}_{x \sim \mathcal{D}}[\ell(h(x), a(x))]$. We introduce a commonly used property of Rademacher complexity:

**Lemma C.2.** *(Property of Rademacher complexity (Mohri et al., 2018)). For any $\delta \in (0, 1)$, with probability at least $1 - \delta$ over the draw of an i.i.d. samples $D$ of size $|D|$, the following inequality*

*holds for all $h \in \mathcal{H}$:*

$$|R_D^\ell(h) - \hat{R}_D^\ell(h)| \leq 2\mathfrak{R}_D(\ell \circ \mathcal{H}) + \sqrt{\frac{\log \frac{1}{\delta}}{2|D|}} \tag{9}$$

where $\mathfrak{R}_D(\ell \circ \mathcal{H})$ is the Rademacher complexity of the function class $\ell \circ \mathcal{H}$ given data $D$.

## D  DIVERGENCES

We include some popular $f$-divergences in Table 8.

Table 8: Popular $f$-divergences and their conjugate functions.

| Divergence | $f(x)$ | Conjugate $f^*(t)$ | $f'(1)$ | Activation func. |
|---|---|---|---|---|
| Kullback-Leibler (KL) | $x \log x$ | $\exp(t-1)$ | 1 | $x$ |
| Reverse KL (KL-rev) | $-\log x$ | $-1 - \log(-t)$ | -1 | $-\exp x$ |
| Jensen-Shannon (JS) | $-x + 1 \log \frac{1+x}{2} + x \log x$ | $-\log(2 - e^t)$ | 0 | $\log \frac{2}{1+\exp(-x)}$ |
| Pearson $\chi^2$ | $(x-1)^2$ | $\frac{t^2}{4} + t$ | 0 | $x$ |
| Total Variation (TV) | $\frac{1}{2}|x-1|$ | $1_{-1/2 \leq t \leq 1/2}$ | $[-1/2, 1/2]$ | $\frac{1}{2}\tanh x$ |

## E  PROOFS

### E.1  PROOF OF LEMMA 3.1

Let us divide the data into four subgroups: unprotected normal examples (UN), protected normal examples (PN), unprotected anomalies (UA), and protected anomalies (PA). For subgroup $t \in \{$UN, PN, UA, PA$\}$, let $\mathcal{L}_0^t$ denote the loss of the unfitted model on $t$, and $\mathcal{L}_1^t$ denote the loss of the fitted model on $t$. $\Delta^t = \mathcal{L}_0^t - \mathcal{L}_1^t > 0$ means the difference of loss between the fitted model and the unfitted one on the subgroup $t$. Assuming that the model can only fit two sets of data, to ensure that the model fits the sets of protected normal examples and unprotected normal examples, we need the following 5 inequalities to hold:

$$(1 - \epsilon)(\mathcal{L}_1^{UN} + \mathcal{L}_0^{UA}) + \epsilon(\mathcal{L}_1^{PN} + \mathcal{L}_0^{PA}) <$$

(1) $(1 - \epsilon)(\mathcal{L}_0^{UN} + \mathcal{L}_1^{UA}) + \epsilon(\mathcal{L}_1^{PN} + \mathcal{L}_0^{PA})$, implied by $\Delta^{UN} > \Delta^{UA}$ which naturally holds;

(2) $(1 - \epsilon)(\mathcal{L}_1^{UN} + \mathcal{L}_0^{UA}) + \epsilon(\mathcal{L}_0^{PN} + \mathcal{L}_1^{PA})$, implied by $\Delta^{PN} > \Delta^{PA}$ which naturally holds;

(3) $(1 - \epsilon)(\mathcal{L}_0^{UN} + \mathcal{L}_1^{UA}) + \epsilon(\mathcal{L}_0^{PN} + \mathcal{L}_1^{PA})$, this case is equivalent to case (1) plus case (2);

(4) $(1 - \epsilon)(\mathcal{L}_1^{UN} + \mathcal{L}_1^{UA}) + \epsilon(\mathcal{L}_0^{PN} + \mathcal{L}_0^{PA})$, we need $\epsilon > \frac{\Delta^{UA}}{\Delta^{UA}+\Delta^{PN}}$;

(5) $(1 - \epsilon)(\mathcal{L}_0^{UN} + \mathcal{L}_0^{UA}) + \epsilon(\mathcal{L}_1^{PN} + \mathcal{L}_1^{PA})$, we need $\epsilon < \frac{\Delta^{UN}}{\Delta^{UN}+\Delta^{PA}}$.

Inequalities (1)(2)(3) are implied by our assumptions regardless of $\epsilon$ and inequalities (4)(5) give the range of $\epsilon$ that $\frac{\Delta^{UA}}{\Delta^{UA}+\Delta^{PN}} < \epsilon < \frac{\Delta^{UN}}{\Delta^{UN}+\Delta^{PA}}$. We design $\epsilon = \frac{\mathcal{L}_0^U - \mathcal{L}_U}{\mathcal{L}_0^U - \mathcal{L}_U + \mathcal{L}_0^P - \mathcal{L}_P}$, and we discuss the following three cases:

- If $\mathcal{L}_U = \mathcal{L}_1^{UN} + \mathcal{L}_0^{UA}, \mathcal{L}_P = \mathcal{L}_1^{PN} + \mathcal{L}_0^{PA}$, then $\epsilon = \frac{\Delta^{UN}}{\Delta^{UN}+\Delta^{PN}}$, which is within the range;
- If $\mathcal{L}_U = \mathcal{L}_1^{UN} + \mathcal{L}_1^{UA}, \mathcal{L}_P = \mathcal{L}_0^{PN} + \mathcal{L}_0^{PA}$, then $\epsilon = 1$, it encourages to fit $\mathcal{L}_P$;
- If $\mathcal{L}_U = \mathcal{L}_0^{UN} + \mathcal{L}_0^{UA}, \mathcal{L}_P = \mathcal{L}_1^{PN} + \mathcal{L}_1^{PA}$, then $\epsilon = 0$, it encourages to fit $\mathcal{L}_U$.

### E.2  ESTIMATIONS OF $\mathcal{L}_0^U$ AND $\mathcal{L}_0^P$

We estimate $\mathcal{L}_0^U = \sum_{i \in U} \|x_i - \overline{G_U(x)}\|^2$ where $\overline{G_U(x)} = \frac{1}{|U|} \sum_{i \in U} G(x_i)$, and $\mathcal{L}_0^P = \sum_{i \in P} \|x_i - \overline{G_P(x)}\|^2$ where $\overline{G_P(x)} = \frac{1}{|P|} \sum_{i \in P} G(x_i)$. Let us denote this by loss1. We also test different designs of $\mathcal{L}_0^U$ and $\mathcal{L}_0^P$ as follows:

Table 9: Performance of FADIG with different designs of $\mathcal{L}_0^U$ and $\mathcal{L}_0^P$.

| Methods | MNIST-USPS (K=1200) | | | MNIST-Invert (K=500) | | |
|---|---|---|---|---|---|---|
| | Recall@K | ROCAUC | Rec Diff | Recall@K | ROCAUC | Rec Diff |
| loss1 | 67.16±0.37 | 91.27±0.49 | 3.73±2.13 | 72.37±0.32 | 98.03±0.01 | 6.75±0.34 |
| loss2 | 66.47±1.73 | 90.60±0.52 | 4.78±2.36 | 72.44±0.74 | 98.04±0.03 | 7.22±0.21 |
| loss3 | 66.31±0.65 | 91.37±0.88 | 6.32±1.74 | 71.39±1.96 | 97.22±1.42 | 8.95±0.92 |
| loss4 | 66.56±2.32 | 90.88±1.67 | 2.54±2.11 | 71.92±3.58 | 97.01±1.85 | 8.96±3.23 |

- loss2: $\mathcal{L}_0^U = \sum_{i \in U} \|x_i\|^2$ and $\mathcal{L}_0^P = \sum_{i \in P} \|x_i\|^2$

- loss3: $\mathcal{L}_0^U = \sum_{i \in U} \|G(x_i) - \overline{x}\|^2$ and $\mathcal{L}_0^P = \sum_{i \in P} \|G(x_i) - \overline{x}\|^2$

- loss4: $\mathcal{L}_0^U = \sum_{i \in U} \|x_i - \overline{x}\|^2$ and $\mathcal{L}_0^P = \sum_{i \in P} \|x_i - \overline{x}\|^2$

We provide results on real-world datasets with these different estimation designs in Table 9. We can see that although the results may vary with different estimation designs, our method always performs better than the baselines in both task performance and fairness.

### E.3 PROOF OF THEOREM 3.3

Given that $D_f(P \parallel Q)$ involves the supremum over all measurable functions and does not account for the hypothesis class, and that it cannot be estimated from finite examples of arbitrary distributions (Kifer et al., 2004), we further consider a discrepancy which helps relieve these issues based on the variational characterization of $f$-divergence in Equation (7):

**Definition E.1.** ($D_{h,\mathcal{H}}^f$ discrepancy (Acuna et al., 2021)) Let $f^*$ be the Fenchel conjugate of a convex, lower semi-continuous function $f$ that satisfies $f(1) = 0$, and let $\hat{T}$ be a set of measurable functions such that $\hat{T} = \{\ell(h(x), h'(x)) : h, h' \in \mathcal{H}\}$ where $\ell$ is a loss function and $\mathcal{H}$ is the hypothesis space. We define the discrepancy between the two distributions $P$ and $Q$ as:

$$D_{h,\mathcal{H}}^f(P \parallel Q) := \sup_{h' \in \mathcal{H}} |\mathbb{E}_{x \sim P}[\ell(h(x), h'(x))] - \mathbb{E}_{x \sim Q}[f^*(\ell(h(x), h'(x)))]|$$

From the definition we can easily get $D_{h,\mathcal{H}}^f(P \parallel Q) \leq D_f(P \parallel Q)$.

With property of the rademacher complexity in Lemma C.2, we now show that $D_{h,\mathcal{H}}^f$ can be estimated from finite examples:

**Lemma E.2.** Suppose $\ell : \mathcal{Y} \times \mathcal{Y} \to [0, 1]$, $f^*$ is L-Lipschitz continuous, and $[0, 1] \subseteq \text{dom } f^*$. Let $U$ and $P$ be two empirical distributions corresponding to datasets containing $m$ and $n$ data points sampled i.i.d. from $P_U$ and $P_P$, respectively. Let us denote $\mathfrak{R}$ as the Rademacher complexity of a given hypothesis class, and define $\ell \circ \mathcal{H} := \{x \mapsto \ell(h(x), h'(x)) : h, h' \in \mathcal{H}\}$. For any $\delta \in (0, 1)$, with probability at least $1 - \delta$, we have:

$$|D_{h,\mathcal{H}}^f(P_U\|P_P) - D_{h,\mathcal{H}}^f(U\|P)| \leq 2\mathfrak{R}_{P_U}(\ell \circ \mathcal{H}) + 2L\mathfrak{R}_{P_P}(\ell \circ \mathcal{H}) + \sqrt{\frac{\log \frac{1}{\delta}}{2n}} + \sqrt{\frac{\log \frac{1}{\delta}}{2m}} \quad (10)$$

*Proof.*

$$D_{h,\mathcal{H}}^f(P_U\|P_P) - D_{h,\mathcal{H}}^f(U\|P) = \sup_{h'\in\mathcal{H}}\{|R_U^\ell(h,h') - R_P^{f^*\circ\ell}(h,h')|\}$$

$$- \sup_{h'\in\mathcal{H}}\{|\hat{R}_U^\ell(h,h') - \hat{R}_P^{f^*\circ\ell}(h,h')|\}$$

$$\leq \sup_{h'\in\mathcal{H}}\|R_U^\ell(h,h') - R_P^{f^*\circ\ell}(h,h')| - |\hat{R}_U^\ell(h,h') - \hat{R}_P^{f^*\circ\ell}(h,h')\|$$

$$\leq \sup_{h'\in\mathcal{H}} |R_U^\ell(h,h') - R_P^{f^*\circ\ell}(h,h') - \hat{R}_U^\ell(h,h') + \hat{R}_P^{f^*\circ\ell}(h,h')|$$

$$= \sup_{h'\in\mathcal{H}} |R_U^\ell(h,h') - \hat{R}_U^\ell(h,h')| + |R_P^{f^*\circ\ell}(h,h') - \hat{R}_P^{f^*\circ\ell}(h,h')|$$

$$\leq 2\mathfrak{R}_{P_U}(\ell\circ\mathcal{H}) + \sqrt{\frac{\log\frac{1}{\delta}}{2m}} + 2\mathfrak{R}_{P_P}(f^*\circ\ell\circ\mathcal{H}) + \sqrt{\frac{\log\frac{1}{\delta}}{2n}}$$

where the last inequality comes from the property of Rademacher complexity. Similarly, by Lemma 5.7 and Definition 3.2 of Mohri et al. (2018) we have: $\mathfrak{R}_{P_P}(f^*\circ\ell\circ\mathcal{H}) \leq L\mathfrak{R}_{P_P}(\ell\circ\mathcal{H})$, with $f^*\circ\ell\circ\mathcal{H} := \{x\mapsto \phi(\ell(h(x),h'(x))) : h,h'\in\mathcal{H}\}$. □

Lemma E.2 shows that the empirical discrepancy $D_{h,\mathcal{H}}^f$ converges to the true discrepancy, and the gap is bounded by the complexity of the hypothesis class and the number of examples. We then present another theorem:

**Theorem E.3.** *Let $h^*$ be the ideal joint hypothesis, i.e., $h^* = \arg\min_{h\in\mathcal{H}} R_U^\ell(h) + R_P^\ell(h)$. The risk difference between the two groups is upper bounded by:*

$$R_P^\ell(h) - R_U^\ell(h) \leq D_{h,\mathcal{H}}^f(P_U\|P_P) + R_U^\ell(h^*) + R_P^\ell(h^*). \tag{11}$$

*Proof.* First, notice that by definition, $f^*(t) = \sup_{x\in\text{dom}f}(xt - f(x)) \geq t - f(1) = t$. Then we can prove:

$$R_P^\ell(h,a_P) \leq R_P^\ell(h,h^*) + R_P^\ell(h^*,a_P) \qquad\qquad\text{(triangle inequality } \ell\text{)}$$

$$= R_P^\ell(h,h^*) + R_P^\ell(h^*,a_P) - R_U^\ell(h,h^*) + R_U^\ell(h,h^*)$$

$$\leq R_P^{f^*\circ\ell}(h,h^*) - R_U^\ell(h,h^*) + R_U^\ell(h,h^*) + R_P^\ell(h^*,a_P)$$

$$\leq |R_P^{f^*\circ\ell}(h,h^*) - R_U^\ell(h,h^*)| + R_U^\ell(h,h^*) + R_P^\ell(h^*,a_P)$$

$$\leq D_{h,\mathcal{H}}^f(P_U\|P_P) + R_U^\ell(h,h^*) + R_P^\ell(h^*,a_P)$$

$$\leq D_{h,\mathcal{H}}^f(P_U\|P_P) + R_U^\ell(h,a_U) + R_U^\ell(h^*,a_U) + R_P^\ell(h^*,a_P)$$

$$= D_{h,\mathcal{H}}^f(P_U\|P_P) + R_U^\ell(h) + R_U^\ell(h^*) + R_P^\ell(h^*)$$

□

Combining Theorem E.3, Lemma E.2 and the property of Rademacher Complexity, we can easily get:

$$R_P^l(h) - R_U^l(h) \leq D_{h,\mathcal{H}}^f(U\|P)$$

$$+ \hat{R}_U^l(h^*) + 4\mathfrak{R}_U(\ell\circ\mathcal{H}) + 2\sqrt{\frac{\log\frac{1}{\delta}}{2m}}$$

$$+ \hat{R}_P^l(h^*) + 2(L+1)\mathfrak{R}_P(\ell\circ\mathcal{H}) + 2\sqrt{\frac{\log\frac{1}{\delta}}{2n}}$$

Since by definition we have $D_{h,\mathcal{H}}^f(U\|P) \leq D_f(U\|P)$, and for $D_f(U\|P) = \mathrm{TV}(U\|P)$, we have:

$$R_P^l(h) - R_U^l(h) \leq \mathrm{TV}(U\|P)$$

$$+ \hat{R}_U^l(h^*) + 4\mathfrak{R}_U(\ell \circ \mathcal{H}) + 2\sqrt{\frac{\log\frac{1}{\delta}}{2m}} \tag{12}$$

$$+ \hat{R}_P^l(h^*) + 2(L+1)\mathfrak{R}_P(\ell \circ \mathcal{H}) + 2\sqrt{\frac{\log\frac{1}{\delta}}{2n}}.$$

Now we motivate why minimizing the objective $\mathcal{L}_{\mathrm{FAC}}$ leads to small $\mathrm{TV}(U\|P)$. Let $U, P$ be the empirical distributions over the common measurable space $\mathcal{X} := \{z_j^U\}_{j=1}^n \cup \{z_k^P\}_{k=1}^m$ with densities $\hat{p}_U, \hat{p}_P$ that are $c_U, c_P$-Lipschitz with respect to $\ell_2$-norm, respectively. Let $x^* := \arg\min_{x \in \mathcal{X}} |\hat{p}_U(x) - \hat{p}_P(x)|$, $\delta := |\hat{p}_U(x^*) - \hat{p}_P(x^*)|$, and

$$\sigma := \sum_{x \in \mathcal{X}} \|x - x^*\| = \sum_{x \in \mathcal{X}} \sqrt{2 - 2\log\mathrm{sim}(x, x^*)},$$

where the equality is due to law of cosine (and that $\mathrm{sim}$ normalizes $z_j$). We first show how $\mathrm{TV}(U\|P)$ is related to $\delta$ and $\sigma$.

**Lemma E.4.**

$$\mathrm{TV}(U\|P) \leq \frac{1}{2}\left(|\mathcal{X}|\delta + (c_U + c_P)\sigma\right).$$

*Proof.*

$$\mathrm{TV}(U\|P) := \frac{1}{2}\sum_{x \in \mathcal{X}} |\hat{p}_U(x) - \hat{p}_P(x)|$$

$$\leq \frac{1}{2}\sum_{x \in \mathcal{X}} |\hat{p}_U(x) - \hat{p}_U(x^*)| + |\hat{p}_U(x^*) - \hat{p}_P(x^*)| + |\hat{p}_P(x^*) - \hat{p}_P(x)| \quad \text{(triangle inequality)}$$

$$= \frac{1}{2}\left(|\mathcal{X}|\delta + \sum_{x \in \mathcal{X}} |\hat{p}_U(x) - \hat{p}_U(x^*)| + |\hat{p}_P(x) - \hat{p}_P(x^*)|\right)$$

$$\leq \frac{1}{2}\left(|\mathcal{X}|\delta + (c_U + c_P)\sum_{x \in \mathcal{X}} \|x - x^*\|\right) \quad \text{(Lipschitz conditions)}$$

$$= \frac{1}{2}\left(|\mathcal{X}|\delta + (c_U + c_P)\sigma\right).$$

$\square$

Next we motivate why minimizing our objective $\mathcal{L}_{\mathrm{FAC}}$ leads to small $\delta$ and $\sigma$ simultaneously, hence small $\mathrm{TV}(U\|P)$. Recall that our fairness-aware contrastive loss is

$$\mathcal{L}_{\mathrm{FAC}} := \mathcal{L}_{\mathrm{fair}} + \mathcal{L}_{\mathrm{unif}},$$

where

$$\mathcal{L}_{\mathrm{fair}} := -\log\left(\sum_{j \in [n]} \sum_{k \in [m]} \mathrm{sim}\left(z_j^U, z_k^P\right)\right),$$

$$\mathcal{L}_{\mathrm{unif}} := \log\left(\sum_{j \neq k} \mathrm{sim}\left(z_j^U, z_k^U\right) + \sum_{j \neq k} \mathrm{sim}\left(z_j^P, z_k^P\right)\right).$$

Intuitively, minimizing $\mathcal{L}_{\mathrm{FAC}}$ leads to small $\mathcal{L}_{\mathrm{fair}}$ and $\mathcal{L}_{\mathrm{unif}}$ simultaneously, which correspond to large $\mathrm{sim}(z_j^U, z_k^P)$ and small $\mathrm{sim}(z_j^U, z_k^U), \mathrm{sim}(z_j^P, z_k^P)$, which in turn correspond to small $\|z_j^U - $

$z_k^P\|$ and large $\|z_j^U - z_k^U\|, \|z_j^P - z_k^P\|$. Hence it is natural to consider the following surrogate losses

$$\mathcal{L}'_{\text{fair}} := \sum_{j,k\in[n]} \|z_j^U - z_k^P\|,$$

$$\mathcal{L}'_{\text{unif}} := -(\sum_{j\neq k} \|z_j^U - z_k^U\| + \|z_j^P - z_k^P\|).$$

Then it follows immediately that $\sigma \leq \mathcal{L}'_{\text{fair}}$, explaining why minimizing our objective $\mathcal{L}_{\text{FAC}}$ (hence $\mathcal{L}'_{\text{fair}}$) leads to small $\sigma$.

To see that $\delta := |\hat{p}_U(x^*) - \hat{p}_P(x^*)|$ cannot be too large, first consider the extreme case where $\{z_j^U\}_{j=1}^n \cap \{z_k^P\}_{k=1}^n = \emptyset$. Without loss of generality let $\|z_1^U - z_1^P\| = \max_{j,k\in[n]}\|z_j^U - z_k^P\|$. Then adjusting $z_1^U, z_1^P$ to be the unit vector on their angle bisector clearly decreases $\mathcal{L}'_{\text{fair}}$ without affecting $\mathcal{L}'_{\text{unif}}$ by much due to high uniformity within $\{z_j^U\}_{j=1}^n$ and $\{z_k^P\}_{k=1}^n$ respectively. Hence we may assume without loss of generality that $z_1^U = z_1^P = x^*$. Next consider the extreme case where $\hat{p}_U(x^*) = \frac{1}{n}$ and $\hat{p}_P(x^*) = 1$. Then adjusting $z_2^P = \arg\max_{x\neq x^*} \sum_{j\in[n]}\|x - z_j^U\|$ clearly decreases $\mathcal{L}'_{\text{unif}}$ without affecting $\mathcal{L}'_{\text{fair}}$ by much due to high uniformity within $\{z_j^U\}_{j=1}^n$. Hence minimizing our objevie $\mathcal{L}_{\text{FAC}}$ leads to small $\delta := |\hat{p}_U(x^*) - \hat{p}_P(x^*)|$.

### E.4 EXTENSION TO MULTI-VALUE MULTI-GROUP CASES

We can easily extend our method to the multi-value multi-group case by modifying our designed $\mathcal{L}_{\text{FAC}}$ and $\mathcal{L}_{\text{REC}}$. In our loss design, $\mathcal{L}_{\text{REC}}$ is an auto-reweighted reconstruction error, and the weight is in proportion with $\mathcal{L}_0^G - \mathcal{L}_G$, where $G$ denotes a certain group. In the multi-valued, multiple protected attributes case, we can extend the design of $\mathcal{L}_{\text{REC}}$ by weighting each reconstruction error of group $G$ with $\frac{\mathcal{L}_0^G - \mathcal{L}_G}{\sum_g \mathcal{L}_0^g - \mathcal{L}_g}$. Then, in $\mathcal{L}_{\text{FAC}}$, $\mathcal{L}_{\text{fair}}$ encourages the representation similarity between different groups, and we can extend it to the similarity between different combinations of multi-valued, multiple protected attributes. $\mathcal{L}_{\text{unif}}$ encourages the uniformity within each group, and we can add the uniformity term for each group in the multi-valued, multiple group case.

## F ADDITIONAL EXPERIMENTS

### F.1 TRAINING DETAILS AND EXPERIMENTAL SETUP

For the COMPAS dataset, we use a two-layer MLP with hidden units of [32, 32]. For all the other datasets, we use MLP with one hidden layer of dimension 128. For our method, we use the Adam optimizer with learning rate $1e^{-3}$ and we set the hyperparameter $\alpha = 4$ across all the data sets. For the four independent runs, we choose random seeds in [40, 41, 42, 43]. All the experiments were executed using one Tesla V100 SXM2 GPUs, supported by a 12-core CPU operating at 2.2GHz.

For the baselines, we use the suggested hyperparameter settings in their original papers. For FairOD, we follow the instructions for the configurations as proposed by their authors using grid search. For DCFOD, the author studied the hyperparameters $\alpha$ and $\beta$, and we set them as their suggested values. FairSVDD suggested setting $\lambda = 1$ for all datasets. ReContrast suggested setting $\alpha = 1$. Other baselines are tested insensitive to hyperparameters and we set them as the default values in their implementations. For a more comprehensive comparison, we test different hyperparameter settings of the two baseline methods (i.e., ReContrast, FairSVDD) on MNIST-USPS with K=1200. The results are shown in Tables 10 and 11. The experimental results show that our method still outperforms the baselines in terms of both anomaly detection performance and fairness, across all the hyperparameter settings. Notice that determining the hyperparameter for these baseline methods solely based on the experimental results involves data leakage. That is why we follow the default value for these hyperparameters mentioned in these papers to avoid data leakage.

For evaluation, since the task is unsupervised, the train and test sets are the same. Following Shekhar et al. (2021); Zhang & Davidson (2021); Ahmed et al. (2021), to evaluate task performance, we use Recall@K and ROCAUC. For fairness evaluation, considering the imbalance between normal examples and anomalies, we focus on Recall Parity in anomaly detection. Given our score-based anomaly detection framework, we would like to state the mathematical formulation of Recall Parity

Table 10: FairSVDD results with different hyperparameter settings.

| Hyperparameter | Recall@K | ROCAUC | Recall Diff |
|---|---|---|---|
| $\lambda = 1$ | $15.62\pm1.52$ | $58.33\pm1.18$ | $13.75\pm2.56$ |
| $\lambda = 2$ | $15.77\pm0.36$ | $58.62\pm1.32$ | $23.78\pm31.87$ |
| $\lambda = 3$ | $16.86\pm0.88$ | $59.24\pm1.61$ | $34.33\pm17.76$ |
| $\lambda = 4$ | $15.36\pm4.20$ | $59.82\pm2.67$ | $33.33\pm0.87$ |
| $\lambda = 5$ | $15.05\pm0.01$ | $55.52\pm2.03$ | $2.26\pm1.50$ |

Table 11: ReContrast results with different hyperparameter settings.

| Hyperparameter | Recall@K | ROCAUC | Recall Diff |
|---|---|---|---|
| $\alpha = -2$ | $63.29\pm2.33$ | $82.80\pm2.40$ | $9.55\pm2.69$ |
| $\alpha = -1$ | $64.18\pm2.80$ | $84.91\pm3.04$ | $23.81\pm2.79$ |
| $\alpha = 0$ | $64.25\pm4.89$ | $84.37\pm5.86$ | $19.38\pm2.62$ |
| $\alpha = 1$ | $64.29\pm3.18$ | $83.46\pm3.77$ | $41.16\pm5.63$ |
| $\alpha = 2$ | $63.04\pm5.98$ | $70.56\pm7.69$ | $18.31\pm7.65$ |

fairness in anomaly detection as: Let anomaly score for example $x$ be $s(x)$ and let $t_K$ be the anomaly score threshold for top-K selection. Then, the predicted normal examples are the ones with $s(x) < t_K$, and the predicted anomalies are those with $s(x) \geq t_K$. The recall parity in anomaly detection requires that $P(s(x) \geq t_K | x \in U, y = 1) = P(s(x) \geq t_K | x \in P, y = 1)$. We use the absolute value of their difference, *i.e.*, Recall Diff, to evaluate the fairness level.

Our code is at: `https://anonymous.4open.science/r/ICLR12584-69B6`

### F.2 EFFECTIVENESS VALIDATION OF FADIG UNDER DIFFERENT CHOICES OF $K$

For the datasets we use, we approximate $K$ as a number that is close to the exact number of the anomalies in hundreds or fifties. We also conduct experiments on the four datasets with different choices of $K$, and the results are in Table 12 and Table 13. The ROCAUC scores are the same as in the main paper. We can also tell from the tables that accuracy difference is inadequate for measuring group fairness in the imbalanced setting.

For potential users, with knowledge of the dataset (i.e., exact numbers of the anomalies), you can choose the number in hundreds closest to the the exact number of anomalies. If you do not know the exact number, with some domain knowledge, you can set $K$ according to the approximated proportion of anomalies in the dataset. E.g., you can set $K$ as 10% of the total sample number if you expect such an anomaly rate in this dataset.

### F.3 RESULTS OF ADDITIONAL METRICS

To provide a more comprehensive assessment of fairness, we now include additional metrics: precision, precision difference@K, and ROCAUC diff which is not related with K selection. The results on the image and tabular datasets are presented in Tables 14 and 15.

We can observe that FADIG still achieves the highest precision across all the datasets while maintaining the low precision difference and ROCAUC difference.

### F.4 COMPARISON WITH REWEIGHTING HEURISTIC

We compare our designed hyperparameter free and parameter free re-balanced autoencoder with a simple heuristic of setting $\epsilon$ as the scaled majority group size, and the results on the four datasets are shown in Table 16. We can observe that compared with the heuristic weight, our designed learnable $\epsilon$ can effectively better enhance task performance and meanwhile promote fairness.

Table 12: Performance on image datasets with different $K$s.

| Methods | MNIST-USPS (K=1000) | | | MNIST-Invert (K=400) | | |
| --- | --- | --- | --- | --- | --- | --- |
| | Recall@K | Acc Diff | Rec Diff | Recall@K | Acc Diff | Rec Diff |
| FairOD | 10.46±1.16 | 4.35±0.33 | 13.21±1.43 | 6.05±0.21 | 2.70±0.15 | 9.99±1.18 |
| DCFOD | 10.24±0.82 | 4.79±1.12 | 8.40±1.83 | 5.57±1.70 | 2.69±0.37 | 8.78±2.31 |
| FairSVDD | 13.75±1.83 | 5.73±5.64 | 13.49±2.55 | 10.57±0.92 | 5.38±3.12 | 14.25±2.96 |
| MCM | 34.38±0.32 | 29.81±0.84 | 52.46±0.94 | 22.48±0.54 | 8.32±1.10 | 64.37±1.66 |
| NSNMF | 33.56±0.70 | 22.26±0.40 | 65.12±2.36 | 43.91±0.84 | 4.54±0.20 | 55.20±0.92 |
| Recontrast | 45.73±2.74 | 10.59±2.62 | 29.62±2.40 | 52.00±4.86 | 13.81±4.30 | 54.96±13.77 |
| FADIG | 61.60±2.50 | 6.50±0.89 | 7.95±5.94 | 62.28±3.24 | 1.62±1.32 | 7.02±4.48 |

Table 13: Performance on tabular datasets with different $K$s.

| Methods | COMPAS (K=300) | | | CelebA (K=4500) | | |
| --- | --- | --- | --- | --- | --- | --- |
| | Recall@K | Acc Diff | Rec Diff | Recall@K | Acc Diff | Rec Diff |
| FairOD | 14.20±1.83 | 3.92±1.63 | 10.75±0.90 | 7.95±0.21 | 4.94±0.25 | 2.26±1.06 |
| DCFOD | 13.10±1.35 | 3.57±2.29 | 7.23±2.82 | 8.64±0.79 | 4.98±0.40 | 9.24±1.12 |
| FairSVDD | 13.02±1.66 | 3.90±2.43 | 9.45±3.80 | 8.82±0.61 | 2.21±0.40 | 10.22±2.33 |
| MCM | 16.87±1.14 | 4.10±1.98 | 10.17±1.64 | 9.26±0.48 | 7.21±5.98 | 28.69±12.14 |
| NSNMF | 17.29±1.42 | 3.60±1.93 | 33.57±1.22 | 8.90±1.09 | 5.66±0.54 | 40.51±1.54 |
| FADIG | 19.14±2.29 | 9.35±3.00 | 4.75±3.69 | 10.56±1.11 | 13.04±0.30 | 5.10±1.52 |

Table 14: Results of additional metrics on the image datasets.

| Methods | MNIST-USPS (K=1200) | | | MNIST-Invert (K=500) | | |
| --- | --- | --- | --- | --- | --- | --- |
| | Precision@K | Precision Diff | ROCAUC Diff | Precision@K | Precision Diff | ROCAUC Diff |
| FairOD | 12.25±1.10 | 6.35±1.79 | 9.10±4.25 | 7.33±0.66 | 10.89±2.07 | 2.78±1.74 |
| DCFOD | 12.39±0.65 | 8.25±0.62 | 10.17±1.49 | 6.60±0.86 | 2.75±0.90 | 2.69±0.41 |
| FairSVDD | 12.39±1.55 | 26.07±24.50 | 22.34±12.91 | 10.07±1.47 | 10.24±1.25 | 23.33±22.45 |
| MCM | 38.92±0.34 | 34.67±1.98 | 19.68±2.36 | 24.52±1.78 | 19.60±5.13 | 16.95±7.40 |
| NSNMF | 39.19±0.81 | 37.54±2.09 | 19.65±0.93 | 49.60±5.89 | 9.24±0.47 | 20.14±0.74 |
| ReContrast | 65.11±3.03 | 9.91±2.67 | 18.77±13.31 | 68.13±5.47 | 26.25±11.11 | 29.29±1.80 |
| FADIG | 67.00±4.33 | 7.54±1.63 | 9.07±1.89 | 69.27±0.76 | 9.97±3.64 | 0.40±0.14 |

Table 15: Results of additional metrics on the tabular datasets.

| Methods | COMPAS (K=350) | | | CelebA (K=5000) | | |
| --- | --- | --- | --- | --- | --- | --- |
| | Precision@K | Precision Diff | ROCAUC Diff | Precision@K | Precision Diff | ROCAUC Diff |
| FairOD | 17.23±2.20 | 6.76±4.76 | 5.59±2.66 | 10.30±0.22 | 6.01±1.44 | 0.66±0.47 |
| DCFOD | 16.57±1.99 | 5.07±6.09 | 6.14±3.01 | 10.07±0.65 | 6.06±2.62 | 2.14±0.96 |
| FairSVDD | 15.81±3.03 | 9.35±3.97 | 8.84±5.36 | 11.60±1.65 | 7.40±2.59 | 5.73±3.92 |
| MCM | 28.68±1.71 | 7.66±2.04 | 8.85±4.32 | 12.17±0.81 | 6.20±2.67 | 8.25±3.16 |
| NSNMF | 31.43±0.23 | 6.83±3.77 | 10.55±0.98 | 11.31±0.64 | 4.09±0.63 | 1.99±0.13 |
| FADIG | 35.14±0.49 | 3.82±0.96 | 4.49±2.77 | 12.96±0.22 | 6.09±0.37 | 1.43±0.04 |

## F.5 COMPARISON WITH ADDITIONAL BASELINES ON ADDITIONAL DATASETS

We compare our method with three variants of the vanilla auto-encoders, (1) Normalized autoencoder (Yoon et al., 2021), (2) MemAE (Gong et al., 2019) and (3) VFAE (Louizos et al., 2016). Furthermore, we also compare our method with two classical baselines dealing with data imbalance. (4) Separate: An intuitive baseline which trains two separate anomaly detectors for two groups. Since we do not know the ratio of anomalies in protected and unprotected group, during the test time, we set the $K$ for top-K selection in proportion to the group ratio (i.e., set top-K as $K\times$ protected group ratio for protected group model, and set top-K for unprotected group model as $K\times$

Table 16: Comparison of our designed rebalancing strategy with group ratio weighting.

| Datasets | FADIG | | | | Group Ratio Reweighting | | | |
|---|---|---|---|---|---|---|---|---|
| | Recall@K | ROCAUC | Rec Diff | Time(s) | Recall@K | ROCAUC | Rec Diff | Time(s) |
| MNIST-USPS | 67.16±0.37 | 91.27±0.49 | 3.73±2.13 | 122.84 | 62.35±0.10 | 87.61±0.47 | 11.87±3.90 | 75.28 |
| MNIST-Invert | 72.37±0.32 | 98.03±0.01 | 6.75±0.34 | 52.28 | 68.33±0.24 | 89.91±0.02 | 74.22±0.26 | 146.45 |
| COMPAS | 34.43±0.42 | 61.85±0.52 | 5.81±4.36 | 17.94 | 33.42±1.61 | 60.49±4.20 | 5.85±5.75 | 15.95 |
| CelebA | 11.94±0.67 | 59.41±0.58 | 4.66±1.72 | 52.81 | 12.75±0.62 | 57.23±0.25 | 13.54±0.89 | 48.12 |

Table 17: Additional results on image datasets. The best score is marked in bold.

| Methods | MNIST-USPS (K=1200) | | | MNIST-Invert (K=500) | | |
|---|---|---|---|---|---|---|
| | Recall@K | ROCAUC | Rec Diff | Recall@K | ROCAUC | Rec Diff |
| FairOD | 12.35±1.13 | 50.00±0.28 | 11.56±0.64 | 7.52±0.74 | 50.40±0.20 | 8.26±1.27 |
| DCFOD | 12.63±0.33 | 50.09±0.27 | 8.99±0.83 | 6.95±0.91 | 50.54±0.54 | **7.23±2.02** |
| FairSVDD | 15.62±1.52 | 58.33±1.18 | 13.75±2.56 | 12.41±0.76 | 49.67±3.98 | 12.46±2.12 |
| MCM | 39.75±0.23 | 78.80±1.02 | 55.81±0.80 | 25.35±0.56 | 80.96±0.49 | 80.13±1.41 |
| NSNMF | 39.16±0.84 | 65.38±0.58 | 62.90±3.84 | 51.79±0.61 | 74.21±0.34 | 51.07±1.79 |
| Recontrast | 64.29±3.18 | 83.46±3.77 | 41.16±5.63 | 64.22±1.60 | 85.13±5.19 | 56.50±11.23 |
| Normalized | 50.38±1.21 | 89.54±0.20 | 45.81±1.65 | 12.16±3.16 | 54.66±33.11 | 80.74±26.14 |
| MemAE | 31.12±6.69 | 66.07±6.69 | 31.19±14.81 | 27.97±7.73 | 77.27±5.08 | 42.06±41.58 |
| VFAE | 17.43±0.25 | 50.49±0.38 | 23.80±0.34 | 10.99±0.67 | 56.15±1.61 | 10.54±1.74 |
| FADIG | **67.19±0.33** | **91.28±0.46** | **3.77±2.18** | **71.82±0.63** | **97.99±0.07** | 9.78±3.10 |

unprotected group ratio). (5) SMOTE (Chawla et al., 2002), a classical method handling data imbalance for tabular data.

We also add a commonly-used large-scale tabular dataset ACSIncome (Ding et al., 2021). For imbalanced anomaly detection setting, we do random sampling in ACSIncome. The resulting dataset has 64794 unprotected samples where 9580 in them are anomalies, and 13778 protected samples where 1222 in them are anomalies. The sensitive attribute is sex and the label is income.

The results are shown in Tables 17 and 18. We can see that FADIG still outperforms almost all the baselines in both task performance and fairness level.

We further add the comparison on a comprehensive image benchmark VisA (Zou et al., 2022) with an advanced backbone EfficientNet-B0 (Tan & Le, 2019). To fit our group-imbalance anomaly detection setting, we select all of Macaronis 1 examples as the unprotected group which contains 1000 normal examples and 100 anomalies. We randomly sample 20% of Macaronis 2 examples as the protected group, which then contains 200 normal examples and 20 anomalies. We show the results in Table 19. We can observe that FADIG always outperforms all the baselines in both utility and fairness, which further shows the effectiveness of our method.

## F.6 Different Anomaly Types

We extend our experimental setup to analyze how our method performs on different types of anomalies. In MNIST-USPS and MNIST-Invert, the normal samples are digit 0 and the anomalies are the digits 1-9. In COMPAS and CelebA, we use whether the sample is reoffending / attractive or not to define normal and abnormal samples. Compared with the two image datasets, the anomalies in the tabular ones are more clustered. Thus, we sample more clustered anomalies on the image data set MNIST-USPS by selecting only digit 1 as the anomalies with the same anomaly amount. The results are shown in Table 20. We can observe that our FADIG achieves the best recall rate and the second-best ROCAUC score, with a relatively low recall difference. Notably, the baselines with extremely low recall differences are showing "fake" fairness since their task performances are very poor.

Table 18: Additional results on tabular datasets. The best score is marked in bold.

| Methods | COMPAS (K=350) | | | CelebA (K=5000) | | | ACSIncome(K=12000) | | |
|---|---|---|---|---|---|---|---|---|---|
| | Recall@K | ROCAUC | Rec Diff | Recall@K | ROCAUC | Rec Diff | Recall@K | ROCAUC | Rec Diff |
| FairOD | 16.56±2.12 | 50.09±1.28 | 7.97±1.23 | 8.93±0.14 | 49.94±0.12 | **0.68±0.56** | 8.43±0.21 | 46.92±0.21 | 6.52±1.02 |
| DCFOD | 16.08±1.94 | 49.55±1.21 | 9.81±1.76 | 9.66±0.69 | 49.92±0.14 | 7.83±1.26 | 8.68±0.72 | 47.04±0.32 | 9.16±0.84 |
| FairSVDD | 15.33±2.10 | 52.68±5.29 | 11.57±4.06 | 10.19±0.50 | 58.40±1.02 | 10.95±1.93 | 10.37±1.92 | 57.04±4.58 | 8.61±1.33 |
| MCM | 21.10±0.54 | 50.97±0.43 | 6.29±2.66 | 11.03±0.38 | 46.23±3.46 | 26.15±9.31 | 10.97±0.20 | 58.05±0.81 | 13.78±2.21 |
| NSNMF | 22.92±0.32 | 57.97±0.66 | 36.78±1.71 | 10.91±0.54 | 50.45±0.30 | 8.04±1.33 | 11.53±0.46 | 58.91±0.29 | 19.08±5.46 |
| Normalized | 26.37±0.55 | 58.73±0.89 | 22.15±6.36 | 6.39±3.40 | 34.92±23.01 | 9.79±3.46 | 10.12±0.78 | 57.63±1.17 | 20.12±4.38 |
| MemAE | 20.15±2.13 | 55.59±2.89 | 20.10±5.88 | 8.93±1.29 | 52.85±0.41 | 14.25±4.61 | 11.08±0.54 | 58.77±0.32 | 22.16±4.89 |
| VFAE | 17.21±2.42 | 53.71±2.32 | 13.43±3.18 | 8.62±0.07 | 48.11±0.49 | 10.00±0.09 | 9.54±0.82 | 54.41±0.96 | 12.17±1.48 |
| Separate | 15.02±2.03 | 44.48±3.09 | 18.60±6.97 | 8.22±0.50 | 44.27±0.91 | 0.91±0.54 | 8.16±0.34 | 43.21±0.58 | 9.54±1.10 |
| SMOTE | 29.92±2.62 | 60.45±4.18 | 6.96±5.75 | 8.14±0.40 | 45.39±0.35 | 5.07±1.43 | 10.81±0.38 | 56.79±1.66 | 6.72±2.91 |
| FADIG | **34.38±0.36** | **61.45±0.47** | 5.97±4.34 | **11.96±0.49** | **59.43±0.42** | 4.72±1.26 | **12.47±0.41** | **60.52±0.59** | **5.47±3.12** |

Table 19: Performance on the VisA data set with EfficientNet.

| Methods | Recall@120 | ROCAUC | Rec Diff |
|---|---|---|---|
| FairOD | 9.17±1.85 | 55.60±9.39 | 10.83±3.44 |
| FairSVDD | 10.81±1.97 | 58.16±3.66 | 11.21±2.90 |
| MCM | 14.78±0.56 | 63.21±0.68 | 28.72±1.96 |
| NSNMF | 16.11±0.81 | 59.23±0.42 | 19.75±2.29 |
| ReContrast | 16.34±3.58 | 67.04±4.12 | 24.31±8.77 |
| FADIG | 17.50±0.23 | 71.71±0.49 | 6.28±0.52 |

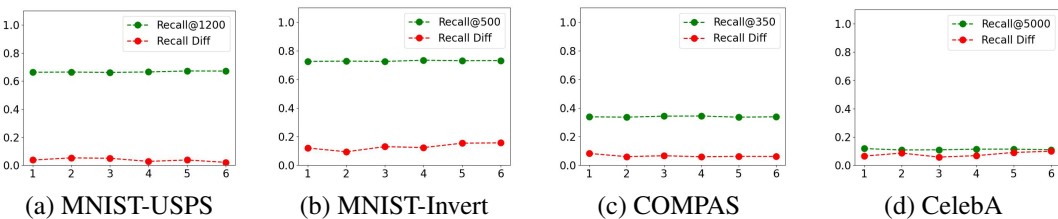

(a) MNIST-USPS     (b) MNIST-Invert     (c) COMPAS     (d) CelebA

Figure 3: Parameter Analysis of $\alpha$ on four datasets. The x-axis is $\alpha$ and the y-axis is for the values of recall and recall difference.

## F.7 GRAPH TASKS

We also compare FADIG on graph tasks with two graph anomaly detection baselines, DOMINANT (Ding et al., 2019) and GRADATE (Duan et al., 2023). We adapt our method on the graph dataset Flickr (Li et al., 2015), replacing the backbone with GCN. The results are shown in Table 21. We can observe that FADIG outperforms DOMINANT in both task performance and fairness. While GRADATE has better task performance compared with our method, it may be because we have not optimized our framework specifically for graph data. In addition, our method achieves a much lower recall difference than both baselines.

## F.8 PARAMETER ANALYSIS

In this section, we conduct the parameter analysis on the four datasets. The experiments are repeated four times and the mean of the recall rate and recall difference are reported. Figure 3 shows the parameter analysis for the parameter $\alpha$ on the four datasets, respectively. The parameter $\alpha$ is used to balance the importance between the reconstruction error and the fair contrastive loss. We can observe that FADIG is robust to the choice of $\alpha$, which may be a benefit from our re-balanced autoencoder.

Table 20: Performance on more clustered anomaly detection.

| Methods | Recall@K | ROCAUC | Rec Diff |
|---|---|---|---|
| FairOD | 12.03±0.42 | 50.04±0.33 | 3.99±3.89 |
| DCFOD | 12.40±1.23 | 49.94±0.72 | 4.08±6.02 |
| FairSVDD | 18.67±1.73 | 54.05±9.09 | 29.72±20.39 |
| MCM | 4.01±0.46 | 14.28±0.60 | 3.81±1.57 |
| NSNMF | 4.64±0.11 | 45.92±0.78 | 12.46±0.62 |
| ReContrast | 16.34±1.78 | 51.82±2.26 | 40.81±18.42 |
| FADIG | 21.04±1.27 | 53.91±1.22 | 14.39±0.43 |

Table 21: Performance on the Flickr dataset.

| Methods | Recall@K | ROCAUC | Rec Diff |
|---|---|---|---|
| DOMINANT | 21.34±0.48 | 61.72±0.59 | 20.56±3.32 |
| GRADATE | 24.96±0.62 | 66.54±1.12 | 35.63±5.34 |
| FADIG | 23.10±0.61 | 63.89±1.12 | 5.33±1.52 |

# G  LIMITATIONS AND BROADER IMPACT

This paper proposes a fairness-aware anomaly detection method, which aims to provide fair results when the algorithm is applied to detect anomalies. Our method currently focus on the binary group fairness case. We can naturally extend our framework to the multi-value multi-group case by extending our design of $\mathcal{L}_{\text{FAC}}$ and $\mathcal{L}_{\text{REC}}$. Incoporating individual fairness notions would be an interesting future direction. By embedding fairness into anomaly detection algorithms, this work contributes to reducing bias and discrimination in AI applications, ensuring that technologies serve diverse populations equitably. In sectors such as finance, healthcare, and law enforcement, where anomaly detection plays a crucial role in identifying fraud, diseases, and criminal activities, incorporating fairness principles can prevent the perpetuation of historical biases and protect vulnerable groups from unjust outcomes. Furthermore, by advancing fairness in AI, this research aligns with global efforts to promote ethics in technology development, fostering trust between AI systems and their users.

