# OpenReview forum: "Rethinking Fair Anomaly Detection From The Group Imbalance Perspective"
_ICLR.cc/2026/Conference — Submitted to ICLR 2026_

### Official Review · Reviewer_TAVv · 2025-10-16

**Soundness:** 3
**Presentation:** 2
**Contribution:** 2
**Rating:** 4
**Confidence:** 4

**Summary:**

This paper introduces FADIG, a new method for fair anomaly detection considering imbalanced protected and unprotected groups. It tackles the issue through two main components: an adaptively re-balanced autoencoder that adjusts group contributions to the loss function and a fairness-aware contrastive learning module to align the data representations across groups.

**Strengths:**

1. The considered problem is essential in fairness studies.

2. The presentation is overall smooth and easy to follow.

3. The experiments are extensive.

**Weaknesses:**

1. In the experiment part, fairness-accuracy trade-off comparisons are missing.

2. The re-balancing loss is a well-established technique. The author mentions that the novelty lies in its hyperparameter-free and parameter-free adaptive weight.  While the proposed formulation in Equation (4) is interesting, the idea of dynamically adjusting weights based on training losses or model performance on different data subsets is not new in the fields of fairness, class imbalance, or hard-sample mining.

3. The paper's motivation hinges on the claim that existing fairness-aware methods "often overlook the underlying group imbalance that gives rise to such unfairness". This framing is confusing, as any method designed to enforce group fairness inherently must consider group imbalance. The limitation is not that prior work overlooks imbalance, but how it attempts to solve it but insufficient.

4. The motivation of the proposed methods is not well presented. The paper does not provide a clear explanation for why FADIG's specific components succeed where others fail. The experimental results show that FADIG outperforms the baselines, and the ablation study confirms its components are necessary. However, it does not show how/why the proposed methods can achieve their superior outcomes by resolving what kinds of limitations.

5. Authors should consider widely adopted fairness metrics such as equalized odds and equalized opportunity for results comparisons.

**Questions:**

1. At the beginning, the model is largely unfitted, meaning reconstruction errors are high and potentially close to the initial estimates. This could make the numerator and denominator small, potentially leading to unstable or erratic behavior.

---

> ### Author Response · Authors · 2025-12-03
>
> We thank the reviewer for the careful reading of our paper and for acknowledging the importance of the problem, the clarity of presentation, and the extensiveness of our experiments. We address each concern below.
>
> `W1: Fairness–accuracy trade-off comparisons are missing.`
>
> **A1:** The fairness–accuracy trade-off is precisely what our evaluation focuses on. Across all datasets (Tables 3–4). Fairness-unaware baselines (MCM, NSNMF, ReContrast) achieve high recall but large recall gaps. Existing fairness-aware methods (FairOD, DCFOD, FairSVDD) achieve small gaps but suffer substantial drops in recall and ROCAUC. These comparisons are the fairness–accuracy trade-off: existing methods lie at one end (high fairness, low accuracy) or the other (high accuracy, low fairness), while FADIG achieves both simultaneously.
>
> `W2: Dynamic reweighting is not new; what is novel about Equation (4)?`
>
> **A2:** We agree that reweighting itself is well-established. Our contribution is not the notion of reweighting but rather:
>
> (1) A theoretically derived, fairness-promoting interval for the optimal weight $\epsilon$. Lemma 3.1 identifies a provably correct range of weights guaranteeing that the model learns both protected and unprotected normal patterns without confusion with anomalies. This type of theoretical grounding is not present in prior reweighting methods for class imbalance or fairness.
>
> (2) A closed-form estimator for $\epsilon$ that is parameter-free and hyperparameter-free. Unlike common reweighting approaches (focal loss, importance weighting, hard-mining, inverse-frequency weights), in our method, no tuning is needed, and no group frequencies or external priors are needed. The weight strictly reflects the current model fit, not static heuristics.
>
> (3) Adaptive reweighting specifically tailored to unsupervised anomaly detection. Our derivation leverages the structure of reconstruction errors and the normal/anomalous decomposition, which does not appear in fairness or imbalance literature.
>
>
> `W3: This framing is confusing; fairness-aware methods do consider group imbalance.`
>
> **A3:** We appreciate this comment and agree that our wording can be improved. Our intended meaning is existing fairness-aware anomaly detection methods incorporate fairness constraints, but they do not explicitly model how group imbalance affects representation learning and reconstruction difficulty, leading to the observed failure modes of overcorrection (mislabeling too many majority samples) or underrepresentation (mislabeling minority samples). FADIG succeeds because it explicitly addresses representation bias induced by imbalance, which is a mechanism that fairness constraints alone do not correct.
>
> `W4: The motivation of the proposed methods is not well presented. `
>
> **A4:** Our method consists of two parts:
>
> (1) Adaptive reweighting corrects biased reconstruction learning. Under group imbalance, the autoencoder overfits the majority patterns and exhibits high reconstruction error on minority normal samples. Our adaptive mechanism shifts training focus precisely toward the group with higher residual error, preventing this bias from forming.
>
> (2) Fairness-aware contrastive learning aligns group representations while maintaining anomaly separability. Prior methods either ignore representation alignment (leading to underfitting of minority patterns) or enforce overly strong invariance (leading to collapsed, non-discriminative embeddings). Our contrastive loss balances cross-group alignment (fairness), and within-group uniformity (preserves anomaly structure).
>
> This dual design is what allows FADIG to improve accuracy and fairness simultaneously, a behavior validated across all datasets.
>
> `W5: Authors should consider widely adopted metrics such as equalized odds or equal opportunity.`
>
> **A5:** We want to clarify that recall disparity is mathematically equivalent to the Equal Opportunity gap, because it compares group-wise true positive rates. Additionally, we report accuracy difference and precision difference, which together capture multiple dimensions of group-wise performance disparity and provide a more comprehensive fairness evaluation.
>
> `Q1: At initialization, reconstruction errors are high; does this make Equation (4) unstable?`
>
> **A6:** We appreciate the reviewer’s concern. In practice, we prevent potential instability during the early stage of training by incorporating a short warmup period before activating the adaptive weighting mechanism. After warmup, the adaptive weight becomes stable and consistently improves both fairness and reconstruction quality, as reflected in our empirical results.

---

### Official Review · Reviewer_ufXC · 2025-10-31

**Soundness:** 3
**Presentation:** 3
**Contribution:** 2
**Rating:** 4
**Confidence:** 3

**Summary:**

This paper studies fairness in unsupervised anomaly detection under group imbalance. Existing fairness-unaware models often misclassify protected-group samples as anomalies, while fairness-aware methods overcompensate and degrade overall performance. To address this, the authors propose FADIG, a fairness-aware anomaly detection method that is inspired by reconstruction-based autoencoders. The authors define a new loss function for reconstruction error that has two components: an adaptively rebalanced autoencoder reconstruction loss that dynamically adjusts group contributions during training, and a fairness-aware contrastive learning loss that aligns the representations of protected and unprotected groups while maintaining within-group diversity. A theoretical analysis based on f-divergence shows that minimizing their contrastive regularization reduces group recall disparity. The authors provide extensive experiments on image, tabular, and graph datasets demonstrating that FADIG achieves higher recall with lower disparity than baselines.

**Strengths:**

- Introduces a novel adaptive reweighting mechanism that balances contributions from protected and unprotected groups, and a fairness-aware contrastive learning module that promotes cross-group alignment and within-group diversity.
- Provides extensive experiments across image, tabular, and graph datasets, showing improvements in both fairness and accuracy (higher recall).

**Weaknesses:**

- The core contribution is relatively modest, as FADIG ultimately modifies the training objective through a reweighted reconstruction loss and additional fairness regularizer.
- The fairness analysis only provides an indirect link between the theoretical risk bounds and the empirical fairness metric (recall disparity), limiting the strength of its claims.

**Questions:**

- The authors say that the training and the test datasets are the same. I don't know how common this is in anomaly detection, but this requires more justification than the task being unsupervised.
- I am quite surprised about the recall statistics of fairness-unaware methods. In general, there is a tradeoff between accuracy and fairness, but the algorithm proposed by the authors achieves higher accuracy and fairness simultaneously. This raises questions about whether the baselines are sufficiently strong or well-tuned, how authors explain FADIG's ability to achieve higher fairness without compromising accuracy.

---

> ### Author Response · Authors · 2025-12-03
>
> Thank you for your invaluable feedback. We would like to address your concerns and provide the response below.
>
> `W1: The core contribution seems modest since FADIG modifies the training objective via reweighting and a fairness regularizer.`
>
> **A1:** We appreciate the chance to clarify the conceptual depth of the method. While FADIG is indeed simple to implement, which we view as an advantage, its design and theoretical properties extend beyond a heuristic objective modification:
>
> (1) Adaptive weight $\epsilon$ is analytically derived, not a tuned hyperparameter. Unlike standard reweighting schemes, $\epsilon$ is fully data-driven and grounded in Lemma 3.1, which identifies a provable range of fairness-promoting weights. The closed-form estimator in Eq. (4) is new to fair anomaly detection and dynamically shifts focus toward whichever group is currently underfitted. This addresses fairness and utility without introducing tuning burden.
>
> (2) Fairness-aware contrastive loss simultaneously encourages cross-group alignment and avoids representation collapse. Prior works on fairness typically treat these two goals separately; the dual-term (alignment + uniformity) structure in Eq. (5) is specifically tailored to anomaly detection where both are necessary.
>
> (3) Theoretical fairness guarantee is derived for an unsupervised setting. To our knowledge, FADIG is the first AD model to derive an f-divergence–based bound that directly controls group risk disparity in an unsupervised reconstruction-based context.
>
> We agree that simplicity is part of the method’s value, but the combination of a theoretically grounded adaptive mechanism and a fairness-regularized representation objective constitutes a meaningful contribution beyond prior formulations.
>
> `W2: The theory only indirectly relates to the empirical fairness metric (recall disparity).`
>
> **A2:** Our theorem provides an upper bound on the risk difference between the protected and unprotected groups: $R^l_P(h) - R^l_U(h)$, under any loss function l satisfying the constraints in Theorem 3.3. This formulation is intentionally general. It applies to any group-wise performance disparity defined in terms of a loss, including recall-based losses, precision-related losses, and other metrics commonly used in anomaly detection. In reconstruction-based AD, anomaly detection decisions are obtained by thresholding reconstruction error. Therefore, a smaller risk difference implies that the reconstruction error distributions for the two groups become more similar. When thresholding these reconstruction errors to identify anomalies, such similarity translates into smaller differences in true positive rates across groups. This directly corresponds to recall disparity.
>
> `Q1: Is it standard for training and testing to use the same dataset in anomaly detection?`
>
> **A3:** Yes, this is the standard practice in unsupervised anomaly detection when no anomaly labels are available. The model is trained to reconstruct the overall data distribution without access to anomaly labels. Anomalies appear in the same dataset but do not influence training because the reconstruction objective naturally fits normal samples and fails to fit outliers. This setup is used in nearly all reconstruction-based AD benchmarks.
>
> `Q2: Fairness-unaware methods usually have better accuracy; why does FADIG improve both fairness and recall? Are baselines well-tuned?`
>
> **A4:** All baselines are implemented using publicly available official or author-released code, and tuned according to recommended hyperparameters. We included a detailed discussion in Appendix F.1. The failure of the baselines lies in that under group imbalance, fairness-unaware methods overfit majority patterns; fairness-aware methods overcorrect and sacrifice overall performance. For the supreme performance of FADIG, the adaptive reweighting improves core reconstruction fidelity, addressing the primary source of error. The fairness-aware contrastive loss further aligns the two groups’ latent spaces but preserves within-group uniformity, which maintains anomaly separability. This two-part mechanism enables FADIG to improve representation quality, along with fairness, which translates into both higher performance and better fairness.

---

### Official Review · Reviewer_bFEz · 2025-11-01

**Soundness:** 4
**Presentation:** 4
**Contribution:** 4
**Rating:** 8
**Confidence:** 4

**Summary:**

The paper focuses on fairness in outlier detection in unsupervised learning specifically addressing imbalance data that naturally arises in presence of minority protected groups. The paper presents a method for addressing representation disparity due to imbalance by proposing a fairness-aware contrative learning criterion and a weighted reconstruction based network module to account for patterns from minority groups. Empirical results show the effectiveness of the proposed method across multiple real-world datasets when compared to exciting fairness-aware methods.

**Strengths:**

1. Rebalancing autoencoder with learnable weight for reconstruction loss is a simple way to encourage learning patterns from minority groups. I like the analytical calculation of \epsilon.
2. Adapting contrastive learning for comparing the groups induced by protected attributes is simple and effective approach.
3. Paper is easy to read and follow.

**Weaknesses:**

1. Appendix G notes that the paper focuses on binary groups. How does the method scale with multi valued multiple protected attribute setup?
2. Paper shows the robustness of choices of hyperparameters. However, a discussion on how initial choice was arrived at would be helpful.

**Questions:**

I do not have specific questions. I reviewed this paper in an earlier cycle, and the authors have significantly updated it.

---

> ### Author Response · Authors · 2025-12-03
>
> We sincerely thank the reviewer for the recognition of our work, and we would like to address your concerns below.
>
> `W1:  How does the method scale with multi valued multiple protected attribute setup?`
>
> **A1:** Thank you for raising this point. Our method naturally extends to both multi-valued groups (e.g., race with more than 2 categories) and multiple protected attributes (e.g., gender + race + age). This extension is discussed in detail in Appendix E.4. Briefly: For the Re-balanced autoencoder, we generalize the adaptive weighting scheme by introducing one adaptive weight per group (or per combination of protected-attribute values). The weights remain parameter-free and automatically normalized, and all theoretical guarantees continue to hold because the derivation of Lemma 3.1 extends to more than two subpopulations. For fairness-aware contrastive learning, the contrastive loss generalizes directly by maximizing similarity across all pairs of different protected groups, and enforcing uniformity within each group, leading to a multi-group representation alignment objective similar to multi-class contrastive formulations. This scaling preserves computational complexity comparable to the binary setting and does not require architectural changes.
>
> `W2: Paper shows the robustness of choices of hyperparameters. However, a discussion on how initial choice was arrived at would be helpful.`
>
> **A2:** For the single hyperparameter $\alpha$, which balances reconstruction and contrastive terms, we followed two principles: keeping reconstruction dominant early in training to stabilize autoencoder learning, and ensuring the contrastive term is strong enough to affect representation learning. All other components (e.g., $\epsilon$) are computed directly from data and require no tuning.

---

### Official Review · Reviewer_Zhba · 2025-11-03

**Soundness:** 2
**Presentation:** 2
**Contribution:** 2
**Rating:** 2
**Confidence:** 4

**Summary:**

The paper studies fairness in unsupervised anomaly detection with group-wise imbalance. The authors argues that standard AD methods skew toward majority patterns and over-flag the protected group as anomalous, while prior fairness-aware methods often over-correct and hurt overall recall. To tackle the group imbalance, the authors propose FADIG, which combines (i) an adaptively re-balanced autoencoder that learns group weights to balance utility and fairness, and (ii) a fairness-aware contrastive learning regularizer that aligns group-wise representations without collapsing anomaly separability. Experimental results across multiple datasets shows that FADIG improves detection performance and group parity over baselines.

**Strengths:**

The proposed FADIG framework elegantly integrates adaptive re-weighting and fair contrastive learning, addressing both representation bias and imbalance without requiring group labels during inference.

The authors derive a provable bound showing that minimizing their contrastive regularizer reduces group-risk differences, lending theoretical support to the fairness claims.

**Weaknesses:**

The overall problem setup lacks clarity: the method assumes full access to sensitive attribute labels during training, while anomaly labels remain unavailable. Is this a common real-world scenario? Specifically, addressing partial or missing sensitive information seems to be more reasonable.

The proposed re-balanced reconstruction loss seems to rely on the strong assumption that the minority group also shares worse performance. In practice, however, this is likely to not hold true, which poses concerns on the applicability of the proposed method.

It remains unclear how the proposed method aligns with conventional fairness notions such as disparate impact or equalized odds. The paper primarily evaluates fairness through group-wise performance gaps in anomaly scores, but does not explicitly examine whether FADIG improves or preserves fairness under these established definitions.

Several claims are misleading or inaccurate. For example, the authors claim "a hyperparameter-free and parameter-free adaptively reweighted autoencoder" in line 481, which is clearly not the case.

It is unclear how the proposed method would perform relative to thresholding-based or post-processing fairness baselines, especially given the superiority of post-processing as suggested in existing work [1].

[1] Cruz, André F., and Moritz Hardt. "Unprocessing seven years of algorithmic fairness." arXiv preprint arXiv:2306.07261 (2023).

**Questions:**

Please refer to the weaknesses.

**Details Of Ethics Concerns:**

N/A.

---

> ### Author Response · Authors · 2025-12-03
>
> We thank the reviewer for the constructive feedback and are glad that you found the integration of adaptive weighting and fair contrastive learning elegant and appreciated the theoretical fairness bound. Below we address each concern in detail.
>
> `W1:  The method assumes access to sensitive attributes during training; is this realistic? Partial or missing sensitive information seems more reasonable.`
>
> **A1:** Access to group labels during training only is standard in the fair anomaly-detection literature, including FairOD, DCFOD, and FairSVDD, all of which we follow in our experimental protocol. In many real-world anomaly-detection domains (e.g., fraud detection, credit risk, recidivism datasets like COMPAS), demographic attributes are available during model development because they are collected for auditing or compliance. For example, health records routinely include race, gender, and age, but often lack confirmed diagnostic labels for many conditions. Criminal recidivism data include demographic attributes, yet the ground truth about whether an individual will reoffend is inherently unknown. Thus, our assumption of access to group labels during training but not anomaly labels is aligned with realistic deployment settings and with established fair AD methodologies.  Importantly, our method does not rely on sensitive labels at inference time, and therefore remains deployable in the same scenarios as these prior works. That said, FADIG can be naturally extended to partial availability (e.g., semi-supervised group labels) because both the reconstruction module and contrastive module operate on minibatches and can incorporate missing-label handling.
>
>
> `W2: The approach assumes minority groups have worse reconstruction performance; this may not always hold.`
>
> **A2:** Our method does not assume that the minority group always yields worse reconstruction. Instead, the adaptive weight $\epsilon$ is data-driven and adjusts dynamically based on current model fit, not on any structural assumption about group difficulty. When a group, either majority or minority, is poorly reconstructed, the corresponding term in Eq. (4) increases, and FADIG automatically allocates more weight to that group. When the groups are well balanced, $\epsilon$ naturally converges toward equal contributions. Thus, the adaptive weight does not encode an assumption, but rather responds to the empirical reconstruction behavior, which we formalize in Lemma 3.1 (guaranteeing that $\epsilon$  stays in the fairness-promoting range) and demonstrate empirically in the data-imbalance study (Tables 5–6).
>
> `W3: The paper does not relate results to disparate impact or equalized odds. `
>
> **A3:** Since disparate impact does not consider the ground-truth information, we report recall diff, accuracy diff and precision diff in this work. Importantly, the recall difference we report **is exactly equivalent to the classical Equal Opportunity definition**, which requires equal true positive rates across groups. Equal opportunity is the most meaningful analogue for anomaly detection, where the primary goal is to ensure that anomalies in minority groups are not disproportionately missed. Moreover, our theoretical result (Theorem 3.3) provides an upper bound on risk difference across groups, which directly corresponds to these group fairness gaps in the unsupervised setting. Thus, our fairness regularizer is theoretically grounded in and practically aligned with established group fairness notions.
>
> `W4: Misleading claims. For example, "a hyperparameter-free and parameter-free adaptively reweighted autoencoder" in line 481 is clearly not the case.`
>
>
> **A4:** We want to clarify that the adaptive weight $\epsilon$ introduces no new trainable parameters, requires no tuning, and is computed directly from data each iteration, unlike reweighting heuristics or fairness multipliers that require external validation. So our adaptively reweighted autoencoder is hyperparameter-free and parameter-free.
>
> `W5: It is unclear how the proposed method would perform relative to thresholding-based or post-processing fairness baselines, especially given the superiority of post-processing as suggested in existing work.`
>
> **A5:** Threshold-adjustment methods (including [1]) operate after a scoring function is learned, optimizing fairness via post-hoc calibration. These approaches require access to anomaly labels during adjustment or at minimum validation labels to tune thresholds. In contrast, unsupervised anomaly detection assumes no anomaly labels are available, making threshold-based post-processing inapplicable under our evaluation protocol. Moreover, post-processing cannot correct representation bias, which originates in upstream feature learning and is the main cause of unfairness under group imbalance. FADIG addresses fairness during representation learning, which is why it improves both ROCAUC and recall parity across all imbalance ratios (Tables 3–6).

---

### Meta-Review · Area_Chair_pnSy · 2025-12-18

**Summary:**

After reading the manuscript, reviewer comments, and authors response, I made my recommendation *reject*. Here are the detailed meta review.

**Research Question**

This paper considers the well-defined unsupervised fair outlier detection problem.

**Motivation**

I agree Reviewer TAVv's comments that *"The paper's motivation hinges on the claim that existing fairness-aware methods often overlook the underlying group imbalance that gives rise to such unfairness. This framing is confusing, as any method designed to enforce group fairness inherently must consider group imbalance. The limitation is not that prior work overlooks imbalance, but how it attempts to solve it but insufficient."*

**Philosophy**

The authors aim to start from the group imbalance perspective to tackle the above challenge.

**Solution**

Technically, the authors propose two components, (i) an adaptively re-balanced autoencoder and (ii) fairness-aware contrastive learning. I can understand the first component is designed for the target challenge. However, I did not see any rationality of the second component.

**Theory**

This paper has two parts of theoretical analyses. The first one is for adaptive weight for within the first component. And the second is the Rademacher complexity for the second component. The analysis is fragmented. Especially, the second part cannot present the whole objective function. Moreover, I did not see any empirical verification in the experiments.

**Experiments**

In general, the experimental part is good. Below are my extra suggestions for further improvements.

1. The authors conduct experiments on four datasets. It would be better to involve more datasets.

2. The authors need to verify the motivation. In another word, the authors need to report the results like Table 1 for all datasets.

3. The in-depth exploration can be more interesting. The parameter analysis in Figure 3 demonstrates the proposed method is insensitive for the alpha value from 1-6. This is a small range. In such a case, it might be set to 1 for simplicity.

**Summary**

The proposed method is not well aligned with the target motivation, especially lacking the rationality of the second component. The theoretical analysis is only for the partial objective. Moreover, I did not see any discussion on the trade-off between fairness and utility. All of these make this paper not self-standing.

**Reviewer Concerns:**

The authors note that Reviewers Zhba, bFEz, and ufXC do not appear to have specific expertise in fairness-oriented outlier detection. I agree with this assessment; therefore, their comments were not given primary consideration in our evaluation.

Reviewer TAVv has concerns on the motivation, tradeoff between fairness and utility. I fully agree.

**Reviewer Scores:**

I do not think the author response well addressed Reviewer TAVv's concerns.

---

### Decision · Program_Chairs · 2026-01-26

Reject